# Cross Entropy versus Label Smoothing: A Neural Collapse Perspective

**Li Guo** [*]                                                                                                 *lg154@nyu.edu*
*New York University Shanghai*

**George Andriopoulos**                                                                               *ga73@nyu.edu*
*New York University Abu Dhabi*

**Zifan Zhao**                                                                                             *zz4330@nyu.edu*
*New York University Shanghai*

**Zixuan Dong**                                                                                           *zd662@nyu.edu*
*New York University Shanghai*

**Shuyang Ling**                                                                                         *sl3635@nyu.edu*
*New York University Shanghai*

**Keith Ross** [*]                                                                                   *keithwross@nyu.edu*
*New York University Abu Dhabi*

**Reviewed on OpenReview:** *https://openreview.net/forum?id=FEo55EIvGI*

## Abstract

Label smoothing is a widely adopted technique to mitigate overfitting in deep neural networks. This paper studies label smoothing from the perspective of Neural Collapse (NC), a powerful empirical and theoretical framework which characterizes model behavior during the terminal phase of training. We first show empirically that models trained with label smoothing converge faster to neural collapse solutions and attain a stronger level of neural collapse compared to those trained with cross-entropy loss. Furthermore, we show that at the same level of NC1, models under label smoothing loss exhibit intensified NC2. These findings provide valuable insights into the impact of label smoothing on model performance and calibration. Then, leveraging the unconstrained feature model, we derive closed-form solutions for the global minimizers under both label smoothing and cross-entropy losses. We show that models trained with label smoothing have a lower conditioning number and, therefore, theoretically converge faster. Our study, combining empirical evidence and theoretical results, not only provides nuanced insights into the differences between label smoothing and cross-entropy losses, but also serves as an example of how the powerful neural collapse framework can be used to improve our understanding of DNNs.

## 1 Introduction

The effectiveness of a deep neural network (DNN) hinges significantly on the choice of the loss function during training. While cross-entropy loss is one of the most popular choices for classification tasks, many alternatives with improved empirical performance have been proposed. Among these, label smoothing (Szegedy et al., 2016; Müller et al., 2019; Lukasik et al., 2020) has emerged as a common technique to enhance the performance of DNNs. Instead of supervising the model training with one-hot key labels, label smoothing preprocesses the

---

*Corresponding authors

label for each sample by taking a convex combination of the one-hot label and a uniform distribution This procedure is generally understood as a means of regularization for improving the model's generalizability.

In this paper, we examine the benefits of label smoothing from the perspective of neural collapse (Papyan et al., 2020), a new and powerful framework for obtaining an improved understanding of DNNs.

## Background and Related Work

Deep neural networks (DNNs) typically consist of a non-linear feature extractor and a linear classification layer. Neural collapse, first observed in (Papyan et al., 2020), occurs in DNNs for classification tasks where the data is balanced and cross-entropy loss is employed. It characterizes the geometric properties of the features produced by the feature extractor and the weight vectors of the classifier during the terminal phase of training (TPT):

- NC1: The learned features of samples from the same class approach their respective class means.

- NC2: The collapsed features from different classes and the classification weight vectors approach the vertices of a simplex equiangular tight frame (ETF).

- NC3: Up to rescaling, the linear classifier weights approach the corresponding class means, giving a self-dual configuration.

Other than cross-entropy loss, neural collapse phenomena have been observed and studied under the mean-squared loss as well (Han et al., 2021; Poggio & Liao, 2020; Zhou et al., 2022a). Furthermore, research has extended neural collapse investigations to scenarios involving imbalanced data (Fang et al., 2021; Hong & Ling, 2023; Thrampoulidis et al., 2022; Dang et al., 2023; Hong & Ling, 2023) and cases with a large number of classes (Jiang et al., 2023).

In addition to being observed empirically, neural collapse has also been proven to arise mathematically. Beginning with Mixon et al. (2020), a series of studies have employed approximation models such as the Unconstrained Feature Models (UFMs) (Mixon et al., 2020) or layer-peeled models (Fang et al., 2021) to provide theoretical evidence for the emergence of neural collapse. These optimization models simplify a DNN by treating the last-layer features as free variables to optimize over, which is justified due to the expressiveness of DNNs (Hornik, 1991; Cybenko, 1989; Lu et al., 2017; Shaham et al., 2018). Theoretical advancements in neural collapse not only enhance our understanding of DNNs but also inspire new techniques to improve their performance in diverse applications, such as imbalanced learning (Xie et al., 2023; Liu et al., 2023), transfer learning (Galanti et al., 2021; Li et al., 2022), and continual learning (Yu et al., 2022; Yang et al., 2023), etc.

Several studies have explored UFMs under various loss functions and regularization strategies (Wojtowytsch et al., 2020; Zhu et al., 2021; Dang et al., 2023; Lu & Steinerberger, 2022; Tirer & Bruna, 2022; Tirer et al., 2023; Yaras et al., 2022; Zhou et al., 2022b). Particularly, Zhou et al.(Zhou et al., 2022b) established that under the UFM model, the global minimizers for a wide range of loss functions, including cross-entropy loss with one-hot labels and smoothed labels (label smoothing), exhibit the idealized neural collapse properties. Additionally, they demonstrated that the UFM model has a benevolent landscape that enables the global minimizer to be effectively attained using iterative algorithms. However, their investigation does not provide insights into why label smoothing consistently outperforms cross-entropy loss, or why label smoothing converges faster during training. In this paper, leveraging the neural collapse framework, we conduct an in-depth investigation of training under these loss functions, aiming to explain the reasons behind the observed superiority of label smoothing over cross-entropy loss with one-hot labels.

## Our Contributions

Our study begins with a comprehensive empirical comparison between cross-entropy loss with one-hot labels (hereafter referred to as cross-entropy loss for simplicity) and label smoothing throughout the training process. Specifically, we carefully study how the last layer features and linear classifiers evolve during training. Our findings are as follows:

1. Compared with cross-entropy loss, models trained with label smoothing exhibit accelerated convergence in terms of training error and neural collapse metrics. Furthermore, they converge to a more pronounced level of NC1 and NC2.

2. Along with accelerated convergence, label smoothing maintains a distinct balance between NC1 and NC2. Notably, as compared with cross-entropy loss, label smoothing results in a more pronounced level of NC2 when reaching a comparable level of NC1. We argue that this phenomenon originates from the implicit inductive bias introduced by label smoothing, which equalizes the logits of all non-target classes and thus promotes the emergence of a simplex ETF structure in both the learned features and classification weight. We posit that the emphasis on NC2 in label smoothing, which promotes maximally separable features between classes, enhances the model's generalization performance. Conversely, an excessive level of NC1 may cause the features to overly specialize in the training data, hindering the model's ability to generalize effectively.

3. Models trained with smoothed labels exhibit improved calibration (Guo et al., 2017) by implicitly regularizing classification weights and features during training. However, if temperature scaling (Guo et al., 2017) is applied as post-processing to counteract the regularization effect, label smoothing can lead to deteriorated model calibration. This is because an excessive level of NC1 under label smoothing can cause the model to be overconfident in its predictions, even when they are incorrect, thereby negatively impacting model calibration.

Of equal importance to the empirical results, we perform a mathematical analysis of the convergence properties of the UFM models under cross-entropy loss and label smoothing. While Zhou et al. (Zhou et al., 2022a) demonstrate that, for a broad class of loss functions, including cross-entropy loss and label smoothing, the global minimizers exhibit neural collapse properties, the authors neither derive the exact form of these global minimizers nor thoroughly examine the landscape around them.

1. We first derive closed-form solutions for the global minimizers under both loss functions, which explicitly depend on the smoothing hyperparameter $\delta$.

2. Utilizing these closed-form solutions, we conduct a second-order theoretical analysis of the optimization landscape around their respective global optimizers. Within the UFM context, our mathematical analysis reveals that label smoothing exhibits a more well-conditioned landscape around the global minimum, which facilitates the faster convergence observed in our empirical study.

This paper provides a significantly deeper understanding of why label smoothing provides better convergence and performance than cross-entropy loss. Additionally, the paper illustrates how the powerful framework of neural collapse and its associated mathematical models can be employed to gain a more nuanced understanding of the "why" of DNNs. However, we acknowledge that theoretical evidence about how neural collapse impacts model generalization is lacking in this study. We hope that our work will inspire future research into the intricate interplay between neural collapse, convergence speed, and model generalizability.

## 2 Preliminaries

### 2.1 The Problem Setup

A deep neural network is comprised of two key components: a feature extractor and a linear classifier. The feature extractor $\phi_{\boldsymbol{\theta}}(\cdot)$ is a nonlinear mapping that maps the input $\boldsymbol{x}$ to the corresponding feature embedding $\boldsymbol{h} := \phi_{\boldsymbol{\theta}}(\boldsymbol{x}) \in \mathbb{R}^d$. Meanwhile, the linear classifier involves a weight matrix $\boldsymbol{W} = [\boldsymbol{w}_1, \boldsymbol{w}_2, \cdots, \boldsymbol{w}_K] \in \mathbb{R}^{d \times K}$ and a bias vector $\boldsymbol{b} \in \mathbb{R}^K$. Consequently, the architecture of a deep neural network is captured by the following equation:

$$f_{\Theta}(\boldsymbol{x}) := \boldsymbol{W}^{\top} \phi_{\boldsymbol{\theta}}(\boldsymbol{x}) + \boldsymbol{b}, \tag{1}$$

where $\Theta := \{\boldsymbol{\theta}, \boldsymbol{W}, \boldsymbol{b}\}$ represents the set of all model parameters. In this work, we consider training a deep neural network using a balanced dataset denoted as $\{(\boldsymbol{x}_{ki}, \boldsymbol{y}_{ki})\}_{1 \leq k \leq K, 1 \leq i \leq n}$. This dataset consists of samples distributed across $K$ distinct classes, with $n$ samples allocated per class. Here, $\boldsymbol{x}_{ki}$ represents the $i$-th sample from the $k$-th class, and $\boldsymbol{y}_{ki}$ is a one-hot vector with unity solely in the $k$-th entry. Our objective is to learn

the parameters $\Theta$ by minimizing the empirical risk over the total $N = nK$ training samples:

$$\min_{\Theta} \frac{1}{N} \sum_{k=1}^{K} \sum_{i=1}^{n} l\left(f_{\Theta}(\boldsymbol{x}_{ki}), \boldsymbol{y}_{ki}\right) + \frac{\lambda}{2} \|\Theta\|_F^2, \tag{2}$$

where $l(\cdot, \cdot)$ denotes the chosen loss function, and $\lambda > 0$ is the regularization parameter (i.e., the weight decay parameter).

## 2.2 Training Losses

To simplify the notation, we use $\boldsymbol{z} = \boldsymbol{W}^{\top} \phi_{\boldsymbol{\theta}}(\boldsymbol{x}) + \boldsymbol{b}$ to represent the network's output logit vector for a given input $\boldsymbol{x}$, and $\boldsymbol{p} = \text{Softmax}(\boldsymbol{z})$ to denote the predicted distribution from the model. The cross-entropy between the target distribution $\boldsymbol{y}$ and the predicted distribution $\boldsymbol{p}$ is defined as $l_{CE}(\boldsymbol{y}, \boldsymbol{p}) = -\sum_k y_k \log(p_k)$, where $\boldsymbol{y}$ is a one-hot vector with a value of 1 in the dimension corresponding to the target class. In contrast, label smoothing minimizes the cross-entropy between the smoothed soft label $\boldsymbol{y}^{\delta}$ and the predicted distribution $\boldsymbol{p}$, denoted as $l_{CE}(\boldsymbol{y}^{\delta}, \boldsymbol{p})$, where the soft label $\boldsymbol{y}^{\delta} = (1 - \delta)\boldsymbol{y} + (\delta/K)\mathbf{1}_K$ combines the hard ground truth label $\boldsymbol{y}$ with a uniform distribution over the labels. Here $\mathbf{1}_K$ denotes the K-dimensional vector of all ones. The hyper-parameter $\delta$ determines the degree of smoothing.

For simplicity, we use the following formulation to represent cross-entropy loss with and without label smoothing:

$$l_{CE}(\boldsymbol{y}^{\delta}, \boldsymbol{p}) = -\sum_k y_k^{\delta} \log(p_k), \tag{3}$$

where $y_k^{\delta} = (1 - \delta)y_k + \delta/K$. The provided loss corresponds to cross-entropy loss with one-hot labels when $\delta = 0$, and for any other value of $\delta \in (0, 1)$, it represents the loss with label smoothing.

# 3 Empirical Analysis of Cross-Entropy and Label Smoothing Losses

This section conducts a comprehensive empirical comparison between cross-entropy loss and label smoothing from the perspective of neural collapse. In Section 3.1, we examine the convergence of the model to neural collapse solutions under cross-entropy loss with and without label smoothing. Our results demonstrate that while models under both loss functions converge to neural collapse, label smoothing induces faster convergence and reaches a more pronounced level of neural collapse. Beyond convergence rates, Section 3.2 delves into the dynamics of convergence, revealing that label smoothing introduces a bias toward solutions with a symmetric simplex ETF structure, thereby enforcing NC2. In Section 3.3, we attempt to understand the impact of label smoothing on model calibration from the perspective of neural collapse. In addition, we also investigate how the smoothing hyperparameter $\delta$ influences neural collapses during the model training in the appendix B.4.

**Experiment Setup**. We conducted experiments on CIFAR-10 (Krizhevsky et al., 2009), CIFAR-100, STL-10 (Coates et al., 2011), and Tiny ImageNet (Deng et al., 2009). Tiny ImageNet is a subset of ImageNet, containing 100,000 samples across 200 classes. To ensure consistency with prior studies (Papyan et al., 2020; Zhu et al., 2021), we use ResNet-18 (He et al., 2016) as the backbone for CIFAR-10 and ResNet-50 (He et al., 2016) for CIFAR-100, STL-10, and Tiny ImageNet.

To isolate the effects of neural collapse and minimize external influences, we apply standard preprocessing without data augmentation. To comprehensively analyze model behavior during TPT, we extend the training period to 800 epochs for CIFAR-10, CIFAR-100, and STL-10, and 300 epochs for Tiny ImageNet. For all datasets, we use a batch size of 128 and train with stochastic gradient descent (SGD) with a momentum of 0.9. The learning rate is initialized at 0.05 and follows a multi-step decay, decreasing by a factor of 0.1 at epochs 100 and 200 for Tiny ImageNet and at epochs 150 and 350 for the other datasets. We use a default weight decay of $5 \times 10^{-4}$, except for the experiments in Section 3.3, where a weight decay value of $1 \times 10^{-4}$ is used.

**Metrics for Measuring NC**. We assess neural collapse in the last-layer features and the classifiers using metrics based on the properties introduced in Section 1, with metrics similar to those presented in (Papyan

et al., 2020). For convenience, we denote the global mean and class mean of the last-layer features as:

$$\boldsymbol{h}_G = \frac{1}{N} \sum_{k=1}^{K} \sum_{i=1}^{n} \boldsymbol{h}_{ki}, \quad \bar{\boldsymbol{h}}_k = \frac{1}{n} \sum_{i=1}^{n} \boldsymbol{h}_{ki}, (1 \le k \le K).$$

**Within class variability (NC1)** measures the relative magnitude of the within-class covariance matrix $\Sigma_W := \frac{1}{N} \sum_{k=1}^{K} \sum_{i=1}^{n} (\boldsymbol{h}_{ki} - \bar{\boldsymbol{h}}_k)(\boldsymbol{h}_{ki} - \bar{\boldsymbol{h}}_k)^\top$ compared to the between-class covariance matrix $\Sigma_B := \frac{1}{K} \sum_{k=1}^{K} (\bar{\boldsymbol{h}}_k - \boldsymbol{h}_G)(\bar{\boldsymbol{h}}_k - \boldsymbol{h}_G)^\top$ of the last-layer features. It is formulated as:

$$NC_1 = \frac{1}{K} trace\left(\Sigma_W \Sigma_B^\dagger\right), \tag{4}$$

where $\Sigma_B^\dagger$ denotes the pseudo inverse of $\Sigma_B$.

**Distance to simplex ETF (NC2)** quantifies the difference between the product of the classifier weight matrix and the centered class mean feature matrix, and a simplex ETF, defined as follows[1]:

$$NC_2 := \left\| \frac{\boldsymbol{W}^\top \overline{\boldsymbol{H}}}{\left\| \boldsymbol{W}^\top \overline{\boldsymbol{H}} \right\|_F} - \frac{1}{\sqrt{K-1}} \left( \boldsymbol{I}_K - \frac{1}{K} \mathbf{1}_K \mathbf{1}_K^\top \right) \right\|_F, \tag{5}$$

where $\overline{\boldsymbol{H}} = [\bar{\boldsymbol{h}}_1 - \boldsymbol{h}_G, \cdots, \bar{\boldsymbol{h}}_K - \boldsymbol{h}_G] \in \mathbb{R}^{d \times K}$ represents centered class mean matrix.

**Self-duality (NC3)** measures the distance between the classifier weight matrix $\boldsymbol{W}$ and the centered class-means $\overline{\boldsymbol{H}}$:

$$NC_3 := \left\| \frac{\boldsymbol{W}}{\|\boldsymbol{W}\|_F} - \frac{\overline{\boldsymbol{H}}}{\|\overline{\boldsymbol{H}}\|_F} \right\|_F. \tag{6}$$

It is evident that when $NC_2$ in (5) and $NC_3$ in (6) both reach zero, the matrices $\boldsymbol{W}$ and $\boldsymbol{H}$ form the same simplex ETF up to some scaling factor. Thus, our definitions of NC1-NC3 capture the same concepts as the definitions in (Papyan et al., 2020) and (Zhu et al., 2021). We say that neural collapse occurs if NC1, NC2, and NC3 collectively approach zero during the Terminal Phase of Training (TPT).

### 3.1 Terminal Phase Training under Label Smoothing and Cross-Entropy Loss

In this section, we investigate the distinct behaviors exhibited by models trained under cross-entropy loss with and without label smoothing during TPT. Our experiments are conducted on the CIFAR-10, CIFAR-100, STL-10, and Tiny ImageNet datasets, using a default label smoothing hyperparameter of $\delta = 0.05$. The training dynamics are visualized in Figure 1.

The leftmost column in Figure 1 illustrates the progression of training and testing errors throughout the training process. As expected, models trained under label smoothing exhibit lower testing errors compared to those under cross-entropy loss for all datasets, showcasing an improvement in model generalizability. Furthermore, models trained with label smoothing exhibit faster convergence for training and testing errors. To facilitate a clearer comparison, Table 1 reports training and test errors at different epochs. Since the network's final layer behaves as a nearest centroid classifier (NCC) at the neural collapse solution, we also include the corresponding training and test errors for the NCC. These results further confirm that models trained with label smoothing exhibit a faster reduction in error rates compared to those trained with cross-entropy loss. Figure 1 additionally presents the three neural collapse metrics for models trained under cross-entropy loss and label smoothing. While the values of NC3, representing the alignment of $\boldsymbol{W}$ and $\boldsymbol{H}$, remain consistently low and comparable under both loss functions, models with label smoothing exhibit faster convergence in both NC1 and NC2 and eventually reach lower levels for both NC1 and NC2. In Section B.3 of the appendix, we also compare the training loss of models trained with cross-entropy loss and label smoothing, demonstrating that label smoothing also accelerates convergence in terms of training error. We provide a theoretical explanation for these empirical findings in Section 4.

---

[1]When the bias $\boldsymbol{b}$ is an all-zero vector or a constant vector, $NC_2 = 0$ implies that the average logit matrix, defined as $\overline{\boldsymbol{Z}} = \boldsymbol{W}^\top \overline{\boldsymbol{H}} + \boldsymbol{b}\mathbf{1}_K^T$, satisfies $\overline{\boldsymbol{Z}} = a\left(\boldsymbol{I} - \frac{1}{K}\mathbf{1}_K\mathbf{1}_K^\top\right)$ for some constant $a$, i.e., $\overline{\boldsymbol{Z}}$ is a simplex ETF.

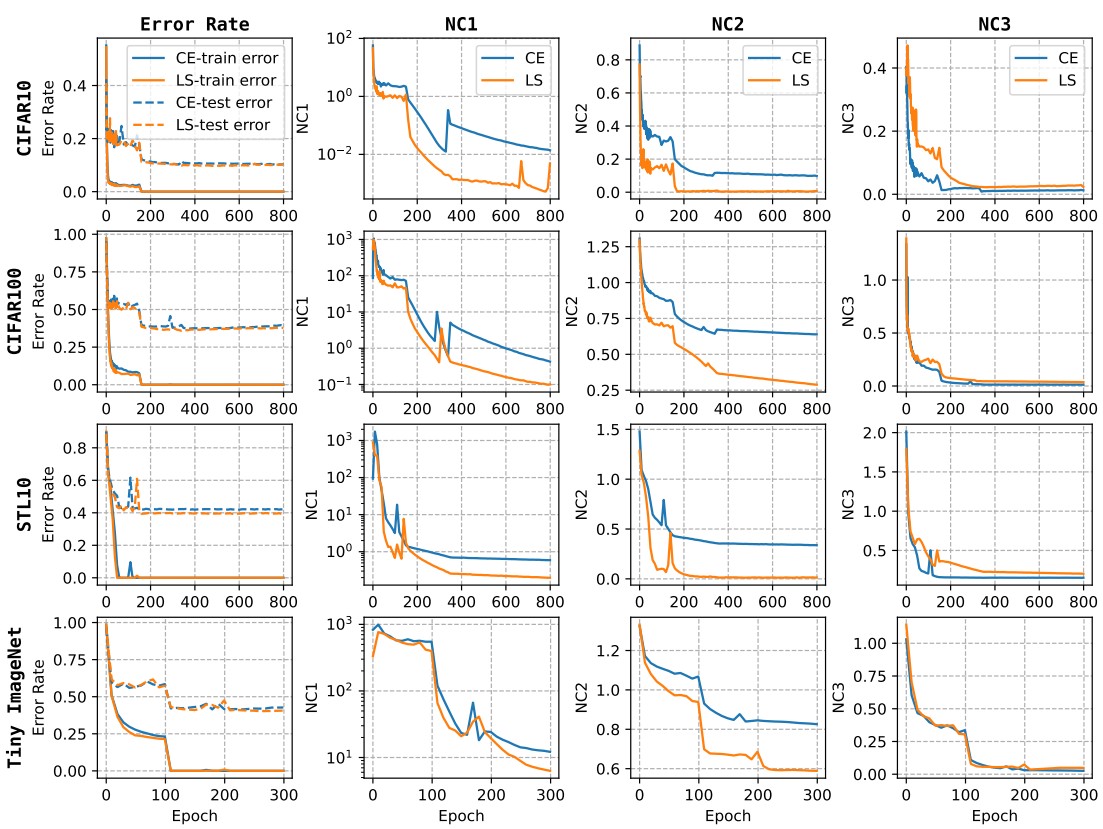

Figure 1: Neural collapse with cross-entropy loss (CE) and label smoothing (LS). Columns from left to right represent the model's error rate (Train/Test), NC1, NC2, and NC3.

## 3.2 Label Smoothing Induces Enhanced NC2

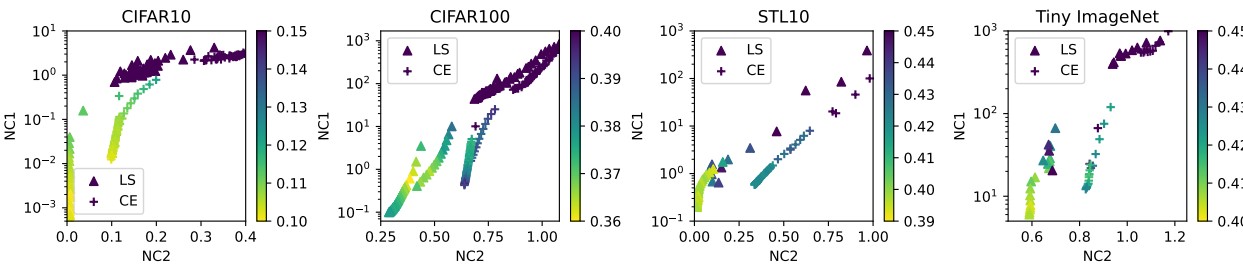

Figure 2: Scatter plots of NC1 vs. NC2 under cross-entropy loss and label smoothing, with colors indicating the test error rate.

Along with the faster convergence to neural collapse under label smoothing, we further observe that models trained with label smoothing maintain a distinct balance between NC1 and NC2 throughout training. Specifically, we find that for the same level of NC1, label smoothing consistently provides a more pronounced manifestation of NC2 compared to cross-entropy Loss. Figure 2 provides scatter plots of NC1 and NC2 under both loss functions, with the NC metrics recorded every 10 training epochs. Across all four datasets, the data points under label smoothing consistently position to the left of those for cross-entropy loss. This observation suggests that at equivalent levels of NC1, label smoothing induces an intensified level of NC2.

| Epoch | Train Error (%) | | Test Error (%) | | NCC Train Error (%) | | NCC Test Error (%) | |
|---|---|---|---|---|---|---|---|---|
| | CE | LS | CE | LS | CE | LS | CE | LS |
| **CIFAR10** | | | | | | | | |
| 10 | 4.84 | 3.41 | 21.23 | 21.60 | 5.51 | 3.53 | 19.67 | 19.28 |
| 20 | 3.59 | 3.15 | 19.67 | 19.34 | 3.06 | 2.49 | 18.05 | 17.26 |
| 50 | 2.67 | 2.28 | 20.68 | 17.52 | 3.75 | 2.39 | 18.79 | 16.86 |
| 100 | 2.27 | 2.07 | 17.49 | 17.26 | 2.10 | 2.47 | 16.90 | 16.60 |
| **CIFAR100** | | | | | | | | |
| 10 | 41.29 | 37.37 | 53.33 | 54.94 | 38.84 | 33.63 | 53.51 | 52.13 |
| 20 | 18.23 | 14.67 | 52.46 | 50.94 | 20.88 | 11.68 | 51.57 | 47.76 |
| 50 | 11.35 | 8.28 | 53.73 | 55.02 | 11.07 | 9.02 | 50.24 | 49.09 |
| 100 | 8.63 | 7.49 | 54.21 | 54.48 | 7.27 | 8.26 | 48.87 | 48.65 |
| **STL10** | | | | | | | | |
| 10 | 65.74 | 62.41 | 67.38 | 65.96 | 67.66 | 64.92 | 68.01 | 65.86 |
| 20 | 55.88 | 47.37 | 58.18 | 53.97 | 64.22 | 52.20 | 66.46 | 56.34 |
| 50 | 0.08 | 0.02 | 52.83 | 40.96 | 12.16 | 0.4 | 50.21 | 41.81 |
| 100 | 0.06 | 0.00 | 45.15 | 42.11 | 0.02 | 0.06 | 44.24 | 40.76 |
| **Tiny ImageNet** | | | | | | | | |
| 10 | 51.04 | 50.67 | 59.64 | 60.74 | 55.84 | 61.76 | 62.55 | 66.95 |
| 20 | 38.30 | 35.59 | 56.47 | 56.79 | 42.17 | 39.85 | 55.86 | 54.55 |
| 50 | 27.75 | 23.14 | 56.63 | 56.18 | 31.21 | 25.54 | 54.21 | 52.36 |
| 100 | 23.14 | 21.45 | 57.51 | 57.42 | 25.03 | 19.94 | 53.48 | 51.54 |

Table 1: Training and test error rates based on the final classification layer (left four columns) and nearest centroid decision rule (right four columns) for models trained with cross-entropy (CE) and label smoothing (LS).

To gain insight into this phenomenon, we now closely examine the formulations of cross-entropy loss and label smoothing. Given an input $x$, the output logit and the predicted distribution are equal to $z = f_\Theta(x)$ and $p = \text{Softmax}(z)$, respectively. As per Equation 3, assuming the observation $x$ is from the $k$-th class, cross-entropy loss is formulated as $l_{CE} = -\log(p_k)$. To minimize cross-entropy loss, the emphasis is solely on making the logit of the target class larger than the logits of non-target classes *without* constraining the logit variation among the non-target classes.

On the other hand, label smoothing loss with smoothing hyperparameter $\delta$ is given by

$$l_{LS} = -\left( \left(1 - \frac{K-1}{K}\delta\right) \log(p_k) + \frac{\delta}{K} \sum_{l \neq k} \log(p_l) \right). \tag{7}$$

According to Jensen's inequality, we have

$$\sum_{l \neq k} \log(p_l) = (K-1) \frac{\sum_{l \neq k} \log(p_l)}{K-1} \leq (K-1) \log\left( \frac{\sum_{l \neq k} p_l}{K-1} \right) = (K-1) \log\left( \frac{1 - p_k}{K-1} \right), \tag{8}$$

with equality achieved only if $p_l = p_{l'}$ (for $l, l' \neq k$). This implies that label smoothing loss reaches its minimum only if the predicted probabilities for non-target classes are all equal. *Thus, label smoothing strengthens the inductive bias towards equalizing the logits of the non-target classes.* This property aligns with the definition of $NC_2$ in (5), explaining why label smoothing loss reinforces NC2. As demonstrated in Section B.1 in the appendix, for deep neural networks with L2 regularization, the convergence of $NC_2$ as defined in (5) to zero indicates that both the classification weight $W$ and class mean features $\overline{H}$ converge to a simplex ETF structure. This ETF structure promotes maximally separable features and classifiers, which are inherently more robust to outliers and can theoretically enhance generalization. However, while NC2 is believed to positively impact generalization, the effect of other factors, such as NC1, complicates this relationship. Empirically, we observe slight improvements in generalization for models trained with label smoothing on CIFAR-10 and CIFAR-100,

and more significant improvements on STL-10. Still, due to the interplay between NC1 and NC2, we cannot conclusively establish a direct causal link between NC2 and generalization performance. Further theoretical and empirical work on how these factors influence generalization remains a promising area for future research.

### 3.3  Label Smoothing and Model Calibration

DNN models often suffer from poor calibration, where the assigned probability values to class labels tend to overestimate the actual likelihood of correctness. This issue arises from the high capacity of DNN models, making them prone to overfitting the training data. Guo et al. (Guo et al., 2017) introduced the expected calibration error (ECE) as a metric for assessing model calibration. Additionally, they proposed temperature scaling as an effective post-processing technique for improving calibration, which involves dividing a network's logits by a scalar $T > 0$ before applying softmax, thereby softening (for $T > 1$) or sharpening (for $T < 1$) the predicted probability to make it more aligned with the true confidence levels.

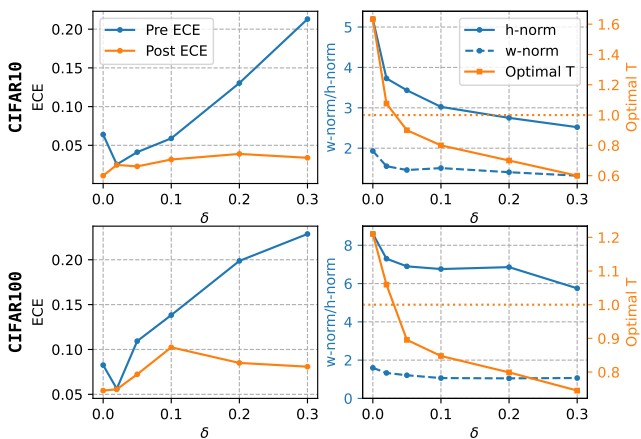

Figure 3: Plots of ECE Pre/Post temperature scaling (left column), alongside feature and classification vector norms and optimal T (right column) for CIFAR10 and CIFAR100 datasets with varying values of $\delta$.

Previous studies (Müller et al., 2019) have highlighted the effectiveness of label smoothing in improving model calibration. Conversely, our empirical study reveals intriguing findings: without temperature scaling, label smoothing (with properly tuned smoothing parameter $\delta$) enhances model calibration compared to cross-entropy loss, *but when temperature scaling is applied as post-processing, label smoothing leads to inferior model calibration.* This inconsistency motivates us to investigate the underlying reason. In this section, we provide an explanation from the perspective of neural collapse.

**Regularization effect of label smoothing on weight and feature norms**. We trained ResNet18 on CIFAR10 and ResNet50 on CIFAR100 using label smoothing with various smoothing hyperparameters ranging from 0 to 0.3. Then, we evaluated models' testing ECE both before and after temperature scaling, as depicted in Figure 3. Our observations reveal that compared to cross-entropy loss ($\delta = 0$), label smoothing with properly chosen $\delta$ lead to lower model calibration error (ECE), while larger values of $\delta$ may degrade model calibration. This is because tuning the smoothing hyperparameter $\delta$ has a similar effect to tuning temperature $T$ in temperature scaling. Particularly, as shown empirically in the right column Figure 3 and theoretically in Theorem 4.1, increasing $\delta$ decreases both the feature and classification vector norms [2], thereby reducing the confidence level of model predictions. Consequently, the corresponding optimal temperature $T$ decreases alongside the increase in $\delta$. In fact, there exists a smoothing hyperparameter $\delta^*$ for which the optimal $T$ equals 1, making temperature scaling post-processing unnecessary. Selecting $\delta > \delta^*$ results in an excessively low confidence level in the model's predictions, leading to a notable increase in test ECE.

However, if we apply temperature scaling to counteract the regularization effect of label smoothing on feature and classification weight, models trained with label smoothing demonstrate higher test ECE (Post ECE as shown in the left column of Figure 3) compared to models trained with cross-entropy loss. We further investigate how this phenomenon is attributed to the more pronounced level of NC1 under label smoothing loss.

---

[2]The feature norm is computed as the average norm of each class mean feature, represented by $\sum_{k=1}^{K} \|\bar{\boldsymbol{h}}_k - \boldsymbol{h}_G\|/K$. The classification vector norm is determined as the mean norm of the weight vector for each class given by $\sum_{k=1}^{K} \|\boldsymbol{w}_k\|/K$, where $\boldsymbol{w}_k \in \mathbb{R}^d$ represents the $k$-th column of the classifier weight $\boldsymbol{W}$.

**Excessive NC1 adversely impacts model calibration**. To investigate the impact of NC1 on model calibration, we analyze how miscalibration occurs during training, using CIFAR10 data with cross-entropy loss and label smoothing ($\delta = 0.05$) as examples. We specifically partition the test set into two subgroups: those correctly classified and those incorrectly classified by the model. In Figure 4, the left column illustrates the average cross-entropy loss and the average entropy of the model predictions for both subgroups of test samples. The middle section of the figure showcases the NC1 metric for the entire test dataset, as well as for each of the subgroups individually. Meanwhile, the right column of Figure 4 presents the training and testing misclassification errors, accompanied by the test set ECE.

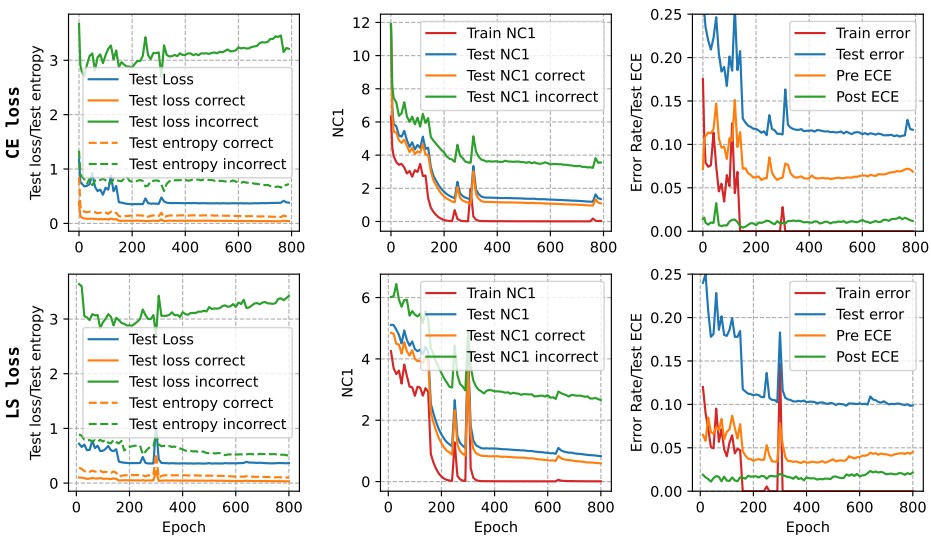

Figure 4: Leftmost plot: average error and entropy for correctly and incorrectly classified test samples. Middle plot: NC1 for the training set, testing set, and correctly/incorrectly classified testing samples. Right plot: model classification error rate and the test ECE before and after temperature scaling.

In Figure 4, models trained with cross-entropy loss and label smoothing show similar trends. For correctly classified test samples, both test loss and average entropy exhibit a consistent decreasing trend, indicating the model's increasing confidence in its correct predictions. Conversely, for incorrectly classified test samples, although the average loss starts to increase after 350 epochs, the overall downward trend of entropy suggests the model's growing confidence in its incorrect predictions. Moreover, the middle plot in Figure 4 indicates that NC1 for both correctly and incorrectly classified test samples consistently decreases during training. For misclassified samples, a smaller NC1 indicates that their feature vectors align more closely with the mean of the incorrectly predicted class, making the model more confident about its incorrect predictions. [3] As a result, the entropy for misclassified test samples continues to decrease, and the test ECE starts to rise during TPT. And if we apply temperature scaling to counteract the regularization effect of label smoothing on feature and classification vector norms, label smoothing loss results in higher ECE due to its stronger NC1.

## 4 Theoretical Analysis

### 4.1 Unconstrained Feature Model

Analyzing deep neural networks poses significant challenges because of the non-linearities and complex interactions between layers in the feature mapping $\boldsymbol{h} := \phi_{\boldsymbol{\theta}}(\boldsymbol{x})$. The Unconstrained Feature Model (UFM) (Fang et al., 2021) simplifies DNN models by treating the features of the last layer as free optimization variables. This choice is motivated by the idea that over-parameterized DNN models are able to approximate any continuous functions (Hornik, 1991; Cybenko, 1989; Lu et al., 2017; Shaham et al., 2018).

---

[3]For misclassified test samples, we assess the within-class variance relative to the incorrectly predicted class.

Recall that the dimension of a feature vector $\boldsymbol{h}$ is denoted by $d$. Let $\boldsymbol{H} = \{\boldsymbol{h}_{ki}\}_{1 \leq k \leq K, 1 \leq i \leq n} \in \mathbb{R}^{d \times N}$ be the feature matrix of all training samples with $\boldsymbol{h}_{ki}$ denoting the $((k-1)n+i)$-th column of $\boldsymbol{H}$. Under the UFM, we investigate regularized empirical risk minimization, a variant of the formulation in Equation 2:

$$\min_{\boldsymbol{W}, \boldsymbol{H}, \boldsymbol{b}} \frac{1}{N} \sum_{k=1}^{K} \sum_{i=1}^{n} l_{CE}(\boldsymbol{W}^\top \boldsymbol{h}_{ki} + \boldsymbol{b}, \boldsymbol{y}_k^\delta) + \frac{\lambda_W}{2} \|\boldsymbol{W}\|_F^2 + \frac{\lambda_H}{2} \|\boldsymbol{H}\|_F^2 + \frac{\lambda_b}{2} \|\boldsymbol{b}\|^2, \tag{9}$$

where $\boldsymbol{y}_k^\delta = (1-\delta)\boldsymbol{e}_k + (\delta/K)\boldsymbol{1}_K$ is the soft label for class $k$ with a smoothing parameter $\delta$. Here $\boldsymbol{e}_k$ a one-hot vector with the $k$-th element equal to 1, and $\lambda_W, \lambda_H, \lambda_b > 0$ are the regularization parameters.

Let $\boldsymbol{Y} = [\boldsymbol{e}_1 \boldsymbol{1}_n^\top, \cdots, \boldsymbol{e}_K \boldsymbol{1}_n^\top] \in \mathbb{R}^{K \times N}$ represent the matrix form of the (hard) ground truth labels. Consequently, the matrix form of the soft labels can be represented as $\boldsymbol{Y}^\delta = (1-\delta)\boldsymbol{Y} + \frac{\delta}{K}\boldsymbol{1}_K \boldsymbol{1}_N^\top$. The empirical risk minimization in (9) has an equivalent matrix form:

$$\min_{\boldsymbol{W}, \boldsymbol{H}, \boldsymbol{b}} \mathcal{L}(\boldsymbol{W}, \boldsymbol{H}, \boldsymbol{b}) := \frac{1}{N} l_{CE}(\boldsymbol{W}^\top \boldsymbol{H} + \boldsymbol{b}\boldsymbol{1}_N^\top, \boldsymbol{Y}^\delta) + \frac{\lambda_W}{2} \|\boldsymbol{W}\|_F^2 + \frac{\lambda_H}{2} \|\boldsymbol{H}\|_F^2 + \frac{\lambda_b}{2} \|\boldsymbol{b}\|^2, \tag{10}$$

where $l_{CE}(\boldsymbol{W}^\top \boldsymbol{H} + \boldsymbol{b}\boldsymbol{1}_N^\top, \boldsymbol{Y}^\delta)$ computes the cross-entropy column-wise and takes the sum.

## 4.2 Theoretical Results

Within the UFM framework, Zhou et al. (Zhou et al., 2022b) demonstrate that, under a wide range of loss functions, including cross-entropy loss and label smoothing, all the global minimizers satisfy neural collapses properties, and in particular $\boldsymbol{H}$ and $\boldsymbol{W}$ form aligned simplex ETFs. Moreover, they show that every critical point is either a global minimizer or a strict saddle point, which implies that these global minimizers can be effectively attained through iterative algorithms. However, their work does not provide the exact expression of the global minimizers, nor does it provide a landscape analysis (condition number) across different loss functions, which is critical for understanding the model's convergence behavior. In contrast, our work derives the exact solutions of both $\boldsymbol{W}$ and $\boldsymbol{H}$ for UFM models with cross-entropy loss and label smoothing. Furthermore, based on these solutions, we employ conditioning number analysis to closely compare the optimization landscape of the models surrounding their respective global minimizers, and thereby provide an explanation for the accelerated convergence observed under label smoothing loss. The proofs of our theorems can be found in the Appendix.

Let $\lambda_Z := \sqrt{\lambda_W \lambda_H}$ and let $a^\delta$ be defined by

(i) if $\sqrt{KN}\lambda_Z + \delta \geq 1$, then $a^\delta = 0$;

(ii) if $\sqrt{KN}\lambda_Z + \delta < 1$, then

$$a^\delta = \log\left(\frac{K}{\sqrt{KN}\lambda_Z + \delta} - K + 1\right).$$

**Theorem 4.1.** *(Global Optimizer). Assume that the feature dimension $d$ is greater than or equal to the number of classes $K$, i.e., $d \geq K$, and the dataset is balanced. Then any global optimizer of $(\boldsymbol{W}, \boldsymbol{H}, \boldsymbol{b})$ of (10) satisfies the following condition:*

*There is a semi-orthogonal matrix $\boldsymbol{P} \in \mathbb{R}^{d \times K}$ (i.e., $\boldsymbol{P}^\top \boldsymbol{P} = \boldsymbol{I}_K$) such that:*

*(i) The classification weight matrix $\boldsymbol{W}$ is given by*

$$\boldsymbol{W} = \left(\frac{n\lambda_H}{\lambda_W}\right)^{1/4} \sqrt{a^\delta} \boldsymbol{P} \left(\boldsymbol{I}_K - \boldsymbol{1}_K \boldsymbol{1}_K^\top / K\right). \tag{11}$$

*(ii) The matrix of last-layer features $\boldsymbol{H}$ is given by*

$$\boldsymbol{H} = \left(\frac{\lambda_W}{n\lambda_H}\right)^{1/4} \sqrt{a^\delta} \boldsymbol{P} \left(\boldsymbol{I}_K - \boldsymbol{1}_K \boldsymbol{1}_K^\top / K\right) \boldsymbol{Y}. \tag{12}$$

*(iii) The bias $\boldsymbol{b}$ is a zero vector, i.e. $\boldsymbol{b} = \boldsymbol{0}$.*

The above theorem provides an explicit closed-form solution for the global minimizer $(\boldsymbol{W}, \boldsymbol{H}, \boldsymbol{b})$. Notably, our findings indicate that an increase in $\delta$ is associated with a decrease in the norm of $\boldsymbol{W}$ and $\boldsymbol{H}$. This observation aligns closely with the empirical results detailed in Section 3.3.

Our observations in Section 3 reveal that label smoothing loss demonstrates accelerated convergence in terms of the NC metrics and training error. To better understand this phenomenon, we analyze the optimization landscape under the UFM. Note that condition number of the Hessian matrix, representing the ratio of the largest to the smallest eigenvalue, plays a crucial role in convergence rate analysis (Nocedal & Wright, 1999; Trefethen & Bau, 2022). In the vicinity of the local minimizer, a smaller condition number typically signifies a faster convergence rate. We now present our main theoretical result, which, to the best of our knowledge, is the first result providing insight into the convergence rate to the optimal solution under UFM models.

At the global optimizer characterized by Theorem 4.1, the predicted probability score for class $k$ can be expressed as [4]

$$\bar{\boldsymbol{p}}_k = (p_t - p_n)\boldsymbol{e}_k + p_n \mathbf{1}_K \tag{13}$$

where $p_t$ and $p_n$ represent the predicted probability for the target and non-target classes, at the global optimal solution. Specifically, these probabilities are defined as:

$$p_t = e^{a^\delta}/(K - 1 + e^{a^\delta}), \quad p_n = 1/(K - 1 + e^{a^\delta}), \tag{14}$$

which implies that $p_t$ decreases as the smoothing parameter $\delta$ increases. With this notation, we state the following theorem.

**Theorem 4.2.** *(Optimization Landscape). Assume the feature dimension d is greater than or equal to the number of classes K ($d \geq K$) and the dataset is balanced. In addition, assume the regularization parameters and the smoothing parameter satisfy $\sqrt{KN\lambda_W \lambda_H} + \delta < 1$. Then, at the global minimizer (as provided in Equation 11-12), the condition number for the Hessian of the empirical loss with respect to $\boldsymbol{W}$ when $\boldsymbol{H}$ is fixed (also the Hessian w.r.t. $\boldsymbol{H}$ when $\boldsymbol{W}$ is fixed) takes the form*

$$\kappa(\nabla_{\boldsymbol{W}}^2 \phi) = \kappa(\nabla_{\boldsymbol{H}}^2 \phi) = K p_t \tag{15}$$

*Therefore, label smoothing with a larger smoothing parameter $\delta$ results in smaller values of $p_t$, which in turn leads to a smaller condition number.*

This theorem suggests that the training with smoothed labels results in a better-conditioned optimization landscape in the vicinity of the global minimizers, facilitating faster convergence behavior. It is noteworthy that while the theorem primarily analyzes the local landscape around the global minimizer, experimental results demonstrate that gradient descent with random initialization indeed converges faster to the global solution of neural collapse under label smoothing loss as observed in Section 3.1 and Section B.6.

One possible intuition for why label smoothing improves the condition number lies in its regularization effect on the predicted probability distribution. Without label smoothing, cross-entropy loss drives the model toward overconfident predictions, assigning nearly all probability mass to the target class and pushing non-target probabilities close to zero. This creates sharp curvature in dimensions corresponding to the target class and flat regions in non-target dimensions, resulting in a poorly conditioned optimization landscape with high curvature variations. Label smoothing mitigates this issue by encouraging a more uniform probability distribution over non-target classes, preventing gradients from being dominated by the target class prediction. By smoothing the loss landscape and reducing curvature differences across dimensions, label smoothing stabilizes gradient updates and promotes more efficient convergence.

However, it is important to note that while Theorem 4.1 establishes a connection between the smoothing parameter $\delta$ and model convergence, it does not directly address how to optimally choose $\delta$ to improve model generalization. This remains an open question for future research.

---

[4]This result follows from the fact that, at the global optimizer, the features of all samples in class $k$ converge to the class mean feature, resulting in identical probability scores as given in Equation 13. Furthermore, due to the simplex equiangular tight frame (ETF) structure of both the class mean features and classifier weight vectors at the global minimizer, we obtain a uniform probability distribution across the non-target classes.

## 5 Discussion

We conducted a comprehensive empirical comparison of cross-entropy loss and label smoothing during training. Our results show that models trained with label smoothing exhibit accelerated convergence, both in terms of training error and neural collapse (NC) metrics. Furthermore, they converge to a more pronounced level of NC1 and NC2. Along with the accelerated convergence, we found that label smoothing maintains a distinct balance between NC1 and NC2. We posit that the emphasis on NC2 in label smoothing enhances the model's generalization performance. Additionally, we investigated the impact of label smoothing on model calibration from the perspective of neural collapse, revealing that label smoothing has a regularization effect on the classifier weight and feature norms, and excessively small NC1 values may adversely affect model calibration.

We performed a mathematical analysis of the convergence properties of the UFM models under CE and LS losses. We first derived closed-form solutions for the global minimizers under both loss functions. Then we conducted a second-order theoretical analysis of the optimization landscape around their respective global optimizers, which reveals that LS exhibits a better-conditioned optimization landscape around the global minimum, which facilitates the faster convergence observed in our empirical study.

This paper provides a significantly deeper understanding of why LS excels in terms of convergence, performance, and model calibration compared to CE loss. Additionally, it illustrates how the powerful framework of neural collapse and its associated mathematical models can be employed to gain a more nuanced understanding of the "why" of DNNs. We expect that these results will inspire future research into the interplay between neural collapse, convergence speed, and model generalizability.

## Acknowledgment

This work is submitted in part by the NYU Abu Dhabi Center for Artificial Intelligence and Robotics, funded by Tamkeen under the Research Institute Award CG010.

This work is partially supported by Shanghai Frontiers Science Center of Artificial Intelligence and Deep Learning at NYU Shanghai. Experimental computation was supported in part through the NYU IT High-Performance Computing resources and services.

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

# A  Appendix

# B  Additional Experiments

This section provides supplementary visualizations to further support the conclusions of the main paper. In Section B.1, we present a more detailed description of the neural collapse metrics, followed by an overview of the experimental setup in Section B.2. Finally, we include additional visualizations to reinforce our key findings.

## B.1  Metrics for Measuring NC

We introduced the metrics for measuring neural collapse in Section 3. For convenience, we restate them here, providing additional details about the subtle differences between our definitions and those used in prior works (Zhu et al., 2021; Zhou et al., 2022b). We denote the global mean and classwise means of the last-layer features as:

$$\boldsymbol{h}_G = \frac{1}{N} \sum_{k=1}^{K} \sum_{i=1}^{n} \boldsymbol{h}_{ki}, \quad \bar{\boldsymbol{h}}_k = \frac{1}{n} \sum_{i=1}^{n} \boldsymbol{h}_{ki}, (1 \le k \le K),$$

where $\boldsymbol{h}_G$ is the global mean and $\bar{\boldsymbol{h}}_k$ represents the mean of class $k$. The metrics for neural collapse are then defined as follows:

**(NC1) Within class variability collapse** measures the relative magnitude of the within-class covariance $\Sigma_W := \frac{1}{N}\sum_{k=1}^{K}\sum_{i=1}^{n}(\boldsymbol{h}_{ki} - \bar{\boldsymbol{h}}_k)(\boldsymbol{h}_{ki} - \bar{\boldsymbol{h}}_k)^\top$ compared to the between-class covariance matrix $\Sigma_B := \frac{1}{K}\sum_{k=1}^{K}(\bar{\boldsymbol{h}}_k - \boldsymbol{h}_G)(\bar{\boldsymbol{h}}_k - \boldsymbol{h}_G)^\top$ of the last-layer features. It is formulated as:

$$NC_1 = \frac{1}{K}trace\left(\Sigma_W \Sigma_B^\dagger\right),$$

where $\Sigma_B^\dagger$ denotes the pseudo inverse of $\Sigma_B$.

**(NC2) Convergence to simplex ETF** quantifies the difference between the normalized classifier weight matrix and the centered class mean features in comparison to a normalized simplex ETF, defined as follows:

$$NC_2 := \left\| \frac{\boldsymbol{W}^\top \overline{\boldsymbol{H}}}{\|\boldsymbol{W}^\top \overline{\boldsymbol{H}}\|_F} - \frac{1}{\sqrt{K-1}}\left(\boldsymbol{I}_K - \frac{1}{K}\boldsymbol{1}_K\boldsymbol{1}_K^\top\right)\right\|_F, \tag{16}$$

where $\overline{\boldsymbol{H}} = [\bar{\boldsymbol{h}}_1 - \boldsymbol{h}_G, \cdots, \bar{\boldsymbol{h}}_K - \boldsymbol{h}_G] \in \mathbb{R}^{d\times K}$ represents centered class mean matrix.

This definition differs slightly from those in (Zhu et al., 2021; Zhou et al., 2022b). However, it can be shown that the convergence of $NC2$ in Equation (16) to zero indicates the simultaneous convergence of both $\boldsymbol{W}$ and $\boldsymbol{H}$ towards the simplex ETF structure. Specifically, when the bias $\boldsymbol{b}$ is an all-zero vector or a constant vector, $NC_2$ as defined in (16) approaching zero indicates that the average logit matrix, formulated as $\overline{\boldsymbol{Z}} = \boldsymbol{W}^\top \overline{\boldsymbol{H}} + \boldsymbol{b}\boldsymbol{1}_K^T$, satisfies the condition $\overline{\boldsymbol{Z}} = a\left(\boldsymbol{I} - \frac{1}{K}\boldsymbol{1}_K\boldsymbol{1}_K^\top\right)$ for some constant $a$. Remarkably, this matrix aligns with the simplex-encoding label (SEL) matrix introduced in (Thrampoulidis et al., 2022), up to a scaling factor $a$.

From Proposition C.4, we have

$$\min_{\boldsymbol{W}^\top \boldsymbol{H} = \boldsymbol{Z}} \frac{1}{2}\left(\lambda_W\|\boldsymbol{W}\|_F^2 + \lambda_H\|\boldsymbol{H}\|_F^2\right) \geq \|\overline{\boldsymbol{Z}}\|_*$$

where $\|\overline{\boldsymbol{Z}}\|_*$ represents the nuclear norm of $\overline{\boldsymbol{Z}}$ and the equality holds only when $\boldsymbol{H} = \overline{\boldsymbol{H}}\boldsymbol{Y}$ and $\overline{\boldsymbol{H}} = \sqrt{\lambda_W/n\lambda_H}\boldsymbol{W}$, indicating the self-duality of the class mean feature and the classification vector. Consequently, during the model training with an L2 penalty on both $\boldsymbol{W}$ and $\boldsymbol{H}$ (with the norm of $\boldsymbol{H}$ implicitly penalized by penalizing the model parameters $\Theta$), the convergence of $NC_2$ as defined in (16) to zero indicates the simultaneous convergence of both $\boldsymbol{W}$ and $\boldsymbol{H}$ towards the simplex ETF structure.

**(NC3) Convergence to self-duality** measures the distance between the classifier weight matrix $\boldsymbol{W}$ and the centered class-means $\overline{\boldsymbol{H}}$:

$$NC_3 := \left\|\frac{\boldsymbol{W}}{\|\boldsymbol{W}\|_F} - \frac{\overline{\boldsymbol{H}}}{\|\overline{\boldsymbol{H}}\|_F}\right\|_F. \tag{17}$$

## B.2 Experiment Setup

In this section, we provide more details for reproducing the experiments presented in the paper. We emphasize that all datasets used—CIFAR-10, CIFAR-100, STL-10, and Tiny ImageNet—are publicly available for academic use. CIFAR-10 and CIFAR-100 are released under the MIT license. All experiments were conducted on a single RTX 3090 GPU with 24GB of memory.

For consistency with prior studies (Papyan et al., 2020; Zhu et al., 2021), we use ResNet-18 (He et al., 2016) as the backbone network for CIFAR-10 and ResNet-50 for CIFAR-100, STL-10, and Tiny ImageNet. To isolate behaviors associated with neural collapse while minimizing the influence of other factors, we apply standard preprocessing techniques without any data augmentation during training. The training period is extended to 300 epochs for Tiny ImageNet and 800 epochs for all other datasets to analyze model behavior during the terminal phase of training. For all datasets, we use a batch size of 128, stochastic gradient descent with a momentum of 0.9, and an initial learning rate of 0.05, which follows a multi-step decay schedule—decreasing by a factor of 0.1 at epochs 100 and 200 for Tiny ImageNet, and at epochs 150 and 350 for the other three datasets. We considered two different scenarios for model regularization:

- **Model with Weight Decay**. For the default setting, we trained the models with weight decay regularization set at $5 \times 10^{-4}$. It's worth noting that the norm of $\boldsymbol{H}$ is implicitly penalized by the weight decay on the model parameters $\Theta$.

- **Simulated UFM Model**. To simulate the unconstrained feature model, we turn off the weight decay and add an L2 penalty on the classifier parameters $\boldsymbol{W}$ and $\boldsymbol{b}$ and the last-layer feature $\boldsymbol{H}$ to the cross-entropy (CE) or label smoothing (LS) loss. This setting was only used in Section B.6 to validate the theoretical results.

## B.3 Comparing Training Loss Under Cross-Entropy and Label Smoothing

From Theorem 4.2, we establish that label smoothing yields a better-conditioned loss landscape, leading to faster convergence. However, due to the different formulations of cross-entropy and label smoothing losses, directly comparing their raw training losses is not meaningful. Instead, in the main paper, we use the training error rate as a proxy and demonstrate that models trained with label smoothing converge faster than those trained with standard cross-entropy loss.

In this section, we present the training loss under both loss functions, as shown in the first row of Figure 5. Since their absolute values are not directly comparable, we also visualize the logarithm of the loss in the second row of Figure 5, defined as $\log(\text{loss} - \min(\text{loss}) + \epsilon)$, where $\min(\text{loss})$ represents the minimum training loss and $\epsilon = 10^{-5}$ is a small constant added to avoid taking the logarithm of zero. As shown in Figure 5, models trained with label smoothing exhibit faster convergence in terms of log-scaled training loss. For a clearer comparison, Table 2 provides a quantitative comparison of the log-scaled loss. Notably, after 150 epochs (or 100 epochs for Tiny ImageNet), the training loss approaches its minimum value, and the log-scaled loss begins to fluctuate significantly. Therefore, we focus our analysis on the loss behavior before this point, where the trend is more stable and informative.

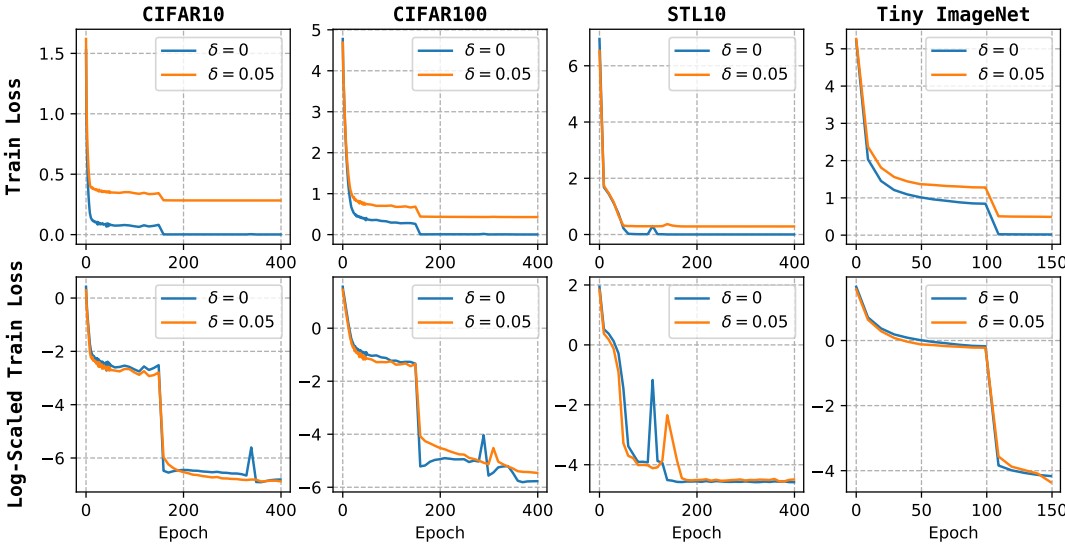

Figure 5: Comparison of training loss (top row) and log-scaled training loss (bottom row) for models trained with cross-entropy loss ($\delta = 0$) and label smoothing ($\delta = 0.05$).

## B.4 Impact of the Smoothing Hyperparameter

In Section 3.1 and 3.2, a default smoothing hyperparameter of $\delta = 0.05$ is utilized. This section explores how the choice of $\delta$ impacts the model's convergence to neural collapse and its generalizability. Specifically, we consider various values for $\delta$ within the interval $[0, 1)$.

| Epoch | CIFAR-10 | | CIFAR-100 | | STL-10 | | Tiny ImageNet | |
|---|---|---|---|---|---|---|---|---|
| | CE | LS | CE | LS | CE | LS | CE | LS |
| 10 | -1.84 | -2.12 | 0.48 | 0.33 | 0.52 | 0.39 | 0.71 | 0.64 |
| 20 | -2.25 | -2.42 | -0.46 | -0.60 | 0.38 | 0.17 | 0.37 | 0.28 |
| 50 | -2.46 | -2.62 | -0.94 | -1.14 | -1.45 | -3.28 | 0.02 | -0.15 |
| 100 | -2.67 | -2.81 | -1.22 | -1.31 | -3.92 | -4.02 | -0.17 | -0.23 |

Table 2: Log-scaled training loss at different epochs for models trained with Cross-Entropy (CE) and Label Smoothing (LS).

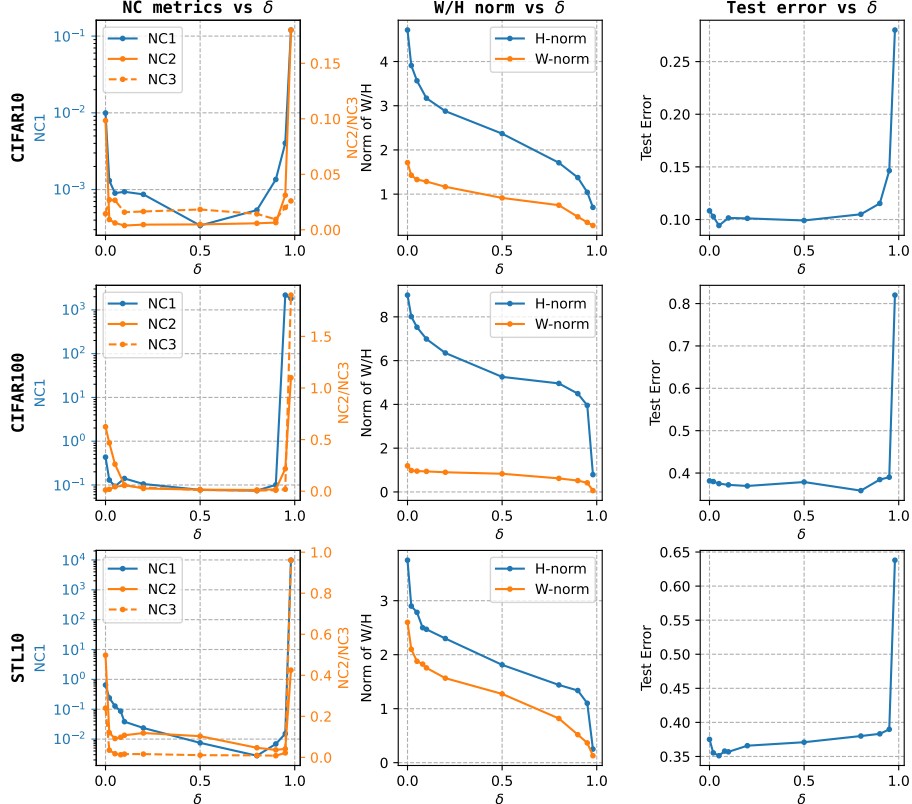

Figure 6: **The effect of the smoothing hyperparameter** $\delta$. The columns from left to right visualize (a) NC metrics vs. $\delta$, (b) Average norm of classification weight vectors and mean features vs. $\delta$, and (c) Testing performance (test error rate) vs. $\delta$.

Figure 6 presents the results of the experiments. The experiments yield several important insights. Firstly, the smoothing hyperparameter significantly influences the model's convergence to neural collapse. The curves of NC1 and NC2 as a function of $\delta$ (2nd column in Figure 6) reveal a U-shaped trend, with the levels of NC3 remaining relatively consistent across different values of $\delta$. Exceptionally small and large values of $\delta$ result in reduced collapse for both NC1 and NC2.

Secondly, the average norms of the classifier vectors and the class mean features decrease as $\delta$ increases. This observation aligns with intuition: given features and classifier vectors that form a simplex ETF structure, decreasing their norms softens the output probabilities. For LS loss, a higher smoothing parameter $\delta$ corresponds to smoother target labels, and consequently, the features and classification vectors with lower norms can achieve close-to-zero label smoothing loss. This observation was also supported by the closed-form

expressions for $\boldsymbol{W}$ and $\boldsymbol{H}$ provided in Theorem 4.1. Thirdly, the trend of the test error also exhibits a U-shape, underscoring the necessity of selecting an appropriate $\delta$. When $\delta$ approaches 1, the nearly uniform smoothed labels do not provide effective training signals to update the parameters. Additionally, the norm of both classification vectors and the last-layer features decrease towards zero, causing the classifier to be dominated by noise, leading to rapid deterioration in model performance. Finally, we observe a surprising robustness in both neural collapse metrics and model performance for the choice of $\delta \in [0.02, 0.8]$. For classification tasks with a larger number of classes, such as CIFAR-100, opting for a relatively large $\delta$ is viable.

## B.5 Weight and Feature Regularization Improve Model Calibration

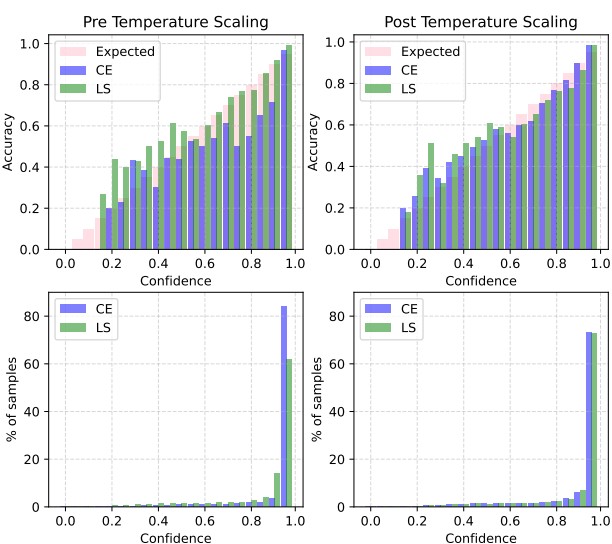

Figure 7: Reliability plots with 20 bins (top row) and the percentage of samples in each bin (bottom row) for models trained with cross-entropy (CE) and label smoothing (LS) losses, both pre- (left column) and post- (right column) temperature scaling. The results are based on the CIFAR-10 dataset.

As mentioned in Section 3.3, label smoothing loss (with properly chosen $\delta$) can improve model calibration. This improvement is due to the regularization effect of label smoothing on the classification vector norm and last-layer feature norm, which leads to predictions with lower confidence. To support this, Figure 7 presents a 20-bin reliability plot (Niculescu-Mizil & Caruana, 2005) for models trained under CE and LS losses for CIFAR-10, with a default smoothing hyperparameter of 0.05. Additionally, we include a plot showing the percentage of samples in each confidence bin. Clearly, LS loss exhibits less confidence in its predictions and better model calibration compared to CE loss without temperature scaling. In the right column of Figure 7, we employ temperature scaling to calibrate the model, with the hyperparameter $T$ selected through cross-validation. Notably, after temperature scaling, models trained with CE loss achieve even better calibration than those trained with LS loss. This is because temperature scaling counteracts the regularization effect of label smoothing on the classification vector norm and feature norm. Consequently, LS loss can result in worse test expected calibration error (ECE) due to an excessive level of NC1.

## B.6 Empirical Validation of Theoretical Results

In Theorem 4.1, we derive closed-form solutions for both the mean feature matrix $\overline{\boldsymbol{H}}$ and the classifier weight matrix $\boldsymbol{W}$, both exhibiting a simplex ETF structure. To validate these theoretical findings, we train ResNet18 models on CIFAR10 dataset with varying smoothing parameters $\delta$. To ensure consistency with the UFM model, we adopted the simulated UFM model as introduced in Section B.2, where weight decay is disabled, and L2 regularization is applied to the last-layer features and classification parameters, ensuring exact regularization effects. We set the regularization parameters as follows: $\lambda_W = 10^{-3}$, $\lambda_H = 10^{-6}$, and

$\lambda_b = 10^{-2}$. Under these settings, we compare the trained average feature norm $\overline{\|\boldsymbol{h}\|} = \sum_{k=1}^{K} \|\bar{\boldsymbol{h}}_k - \boldsymbol{h}_G\|/K$ and average classification vector norm $\overline{\|\boldsymbol{w}\|} = \sum_{k=1}^{K} \|\boldsymbol{w}_k\|/K$ with their theoretical counterparts derived from Theorem 4.1 which can be represented as

$$\overline{\|\boldsymbol{h}\|}^* = \sqrt{(K-1)a^\delta/K} \left( \frac{n\lambda_H}{\lambda_W} \right)^{-1/4}, \quad \overline{\|\boldsymbol{w}\|}^* = \sqrt{(K-1)a^\delta/K} \left( \frac{n\lambda_H}{\lambda_W} \right)^{1/4}.$$

Figure 8 illustrates the differences between the trained predictions $\overline{\|\boldsymbol{h}\|}$ and $\overline{\|\boldsymbol{w}\|}$ and their theoretical solutions for models trained under LS loss with different $\delta$. The plot shows a clear decreasing trend in the relative error between the empirical and theoretical values for both the feature and classification vector norms, with differences reducing to below 0.01 across all cases. These results demonstrate that the empirical findings align with the theoretical predictions, thereby verifying Theorem 4.1.

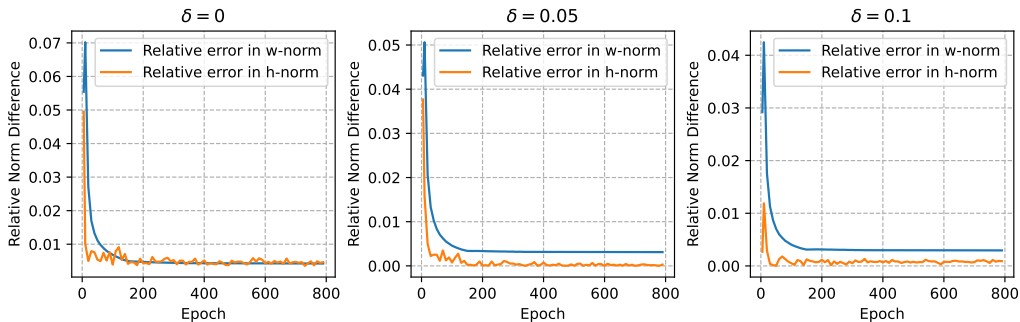

Figure 8: Relative error of classification vector norm $\overline{\|\boldsymbol{w}\|}$ and feature norm $\overline{\|\boldsymbol{h}\|}$ compared to their theoretical results over epochs for ResNet-18 on CIFAR10.

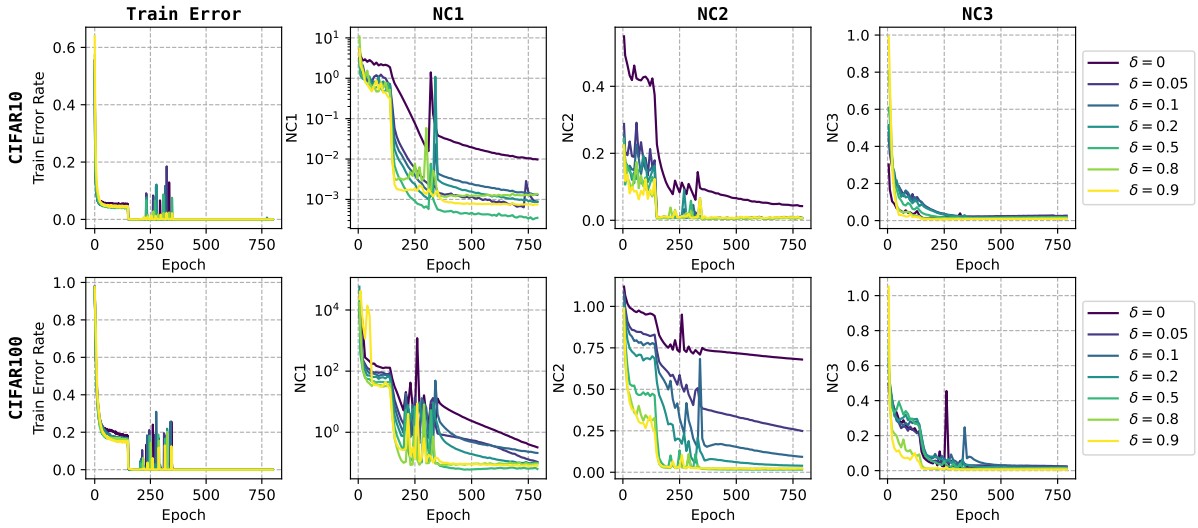

Figure 9: Comparison of model convergence under weight decay regularization. From left to right, the figures display the models' training error rate, NC1, NC2, and NC3 with varying smoothing hyperparameters $\delta$.

According to Theorem 4.2, models trained with label smoothing loss (satisfying $\sqrt{KN}\lambda_Z + \delta < 1$) converges faster than models trained with cross-entropy loss. In Figure 9, we train ResNet18 and ResNet50 on CIFAR10 and CIFAR100, respectively, with a default weight decay value of $5 \times 10^{-4}$. Comparing models trained with different smoothing hyperparameters, we observe that for a wide range of $\delta$ values, models with larger $\delta$ converge faster in terms of both training error and NC metrics.

Furthermore, we investigate model convergence under the simulated UFM model, where weight decay is disabled, and L2 regularization is added. As illustrated in Figure 10, across a wide range of $\delta$ such that $\sqrt{KN}\lambda_Z + \delta < 1$, we observe faster convergence for larger $\delta$ values.

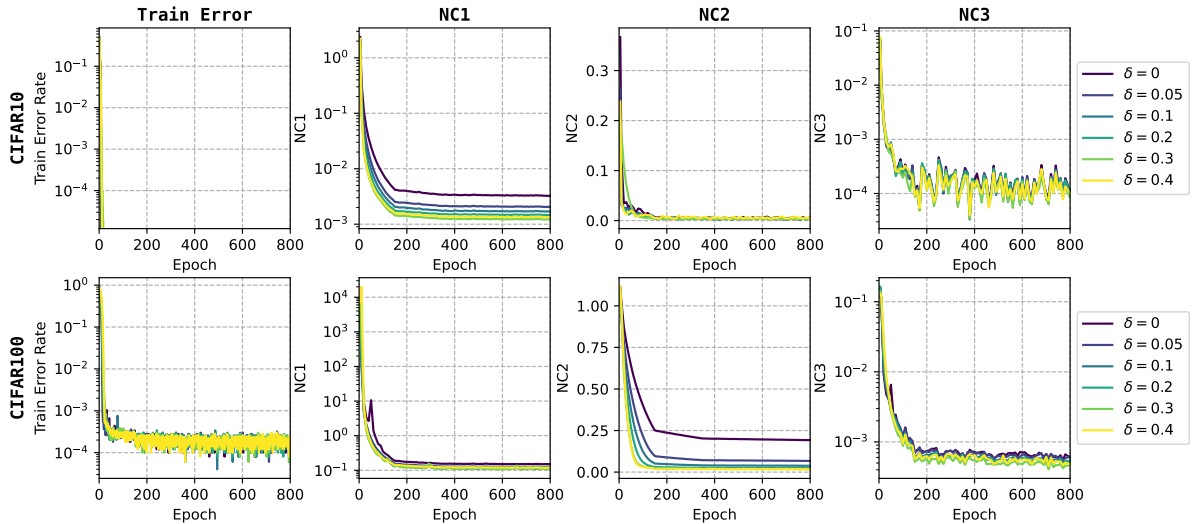

Figure 10: Comparison of model convergence under simulated UFM model. From left to right, the figures display the models' training error rate, NC1, NC2, and NC3 with varying smoothing hyperparameters $\delta$.

## C  Proofs

The first key observation of the phenomenon of neural collapse, that is (NC1), refers to a type of collapse that involves the convergence of the feature of samples from the same class to a unique mean feature vector. The second key observation of neural collapse, namely (NC2), involves these unique mean feature vectors (after recentering by their global mean) as they form an equiangular tight frame (ETF), i.e., they share the same pairwise angles and length. Before providing the theoretical poof, we first formally define the rank $K$ canonical simplex ETF in the definition below.

### C.1  Basics

**Definition C.1.** (K-simplex ETF) A K-simplex ETF is a collection of points in $\mathbb{R}^d$ specified by the columns of the matrix

$$\boldsymbol{M} = \sqrt{\frac{K}{K-1}}\boldsymbol{P}\left(\boldsymbol{I}_K - \frac{1}{K}\mathbf{1}_K\mathbf{1}_K^\top\right),$$

where $\boldsymbol{I}_K \in \mathbb{R}^{K \times K}$ is the identity matrix and $\mathbf{1}_K \in \mathbb{R}^K$ is the ones vector, and $\boldsymbol{P} \in \mathbb{R}^{d \times K}(d \geq K)$ is a partial-orthogonal matrix such that $\boldsymbol{P}^\top\boldsymbol{P} = \boldsymbol{I}_K$.

Note that the matrix $\boldsymbol{M}$ satisfies:

$$\boldsymbol{M}^\top\boldsymbol{M} = \frac{K}{K-1}\left(\boldsymbol{I}_K - \frac{1}{K}\mathbf{1}_K\mathbf{1}_K^\top\right).$$

Next, we prove a series of lemmas that will prove crucial upon establishing our main theorems.

**Lemma C.2.** *(Young's Inequality) Let $p, q$ be positive numbers satisfying $\frac{1}{p} + \frac{1}{q} = 1$. Then for any $a, b \in \mathbb{R}$, we have*

$$|ab| \leq \frac{|a|^p}{p} + \frac{|b|^q}{q},$$

where the equality holds if and only if $|a|^p = |b|^q$. The case for $p = q = 2$ is just the AM-GM inequality which is $|ab| \leq \frac{1}{2}\left(a^2 + b^2\right)$, where the equality holds if and only if $|a| = |b|$

**Lemma C.3.** *For any fixed $\boldsymbol{Z} \in \mathbb{R}^{K \times N}$ and $\alpha > 0$, we have*

$$\min_{\boldsymbol{Z} = \boldsymbol{W}^\top \boldsymbol{H}} \frac{1}{2\sqrt{\alpha}}\left(\|\boldsymbol{W}\|_F^2 + \alpha\|\boldsymbol{H}\|_F^2\right) = \|\boldsymbol{Z}\|_*. \tag{18}$$

*Here $\|\boldsymbol{Z}\|_*$ denotes the nuclear norm of $\boldsymbol{Z}$:*

$$\|\boldsymbol{Z}\|_* := \sum_k \sigma_k(\boldsymbol{Z}) = trace(\Sigma), \ with \ \boldsymbol{Z} = \boldsymbol{U}\Sigma\boldsymbol{V}^\top,$$

*where $\{\sigma_k\}_{k=1}^{min(K,N)}$ denote the singular values of $\boldsymbol{Z}$, and $\boldsymbol{Z} = \boldsymbol{U}\Sigma\boldsymbol{V}^\top$ is the singular value decomposition (SVD) of $\boldsymbol{Z}$.*

*Proof.* Let $\boldsymbol{Z} = \boldsymbol{U}\Sigma\boldsymbol{V}^\top$ be the SVD of $\boldsymbol{Z}$. From the fact that $\boldsymbol{U}\boldsymbol{U}^\top = \boldsymbol{I}$, $\boldsymbol{V}\boldsymbol{V}^\top = \boldsymbol{I}$, and $trace\left(\boldsymbol{A}^\top \boldsymbol{A}\right) = \|\boldsymbol{A}\|_F^2$, we have

$$\|\boldsymbol{Z}\|_* = trace(\Sigma) = \frac{1}{2\sqrt{\alpha}}trace\left(\sqrt{\alpha}\boldsymbol{U}^\top \boldsymbol{U}\Sigma\right) + \frac{\sqrt{\alpha}}{2}trace\left(\frac{1}{\sqrt{\alpha}}\Sigma\boldsymbol{V}^\top \boldsymbol{V}\right)$$

$$= \frac{1}{2\sqrt{\alpha}}\left(\left\|\alpha^{1/4}\boldsymbol{U}\Sigma^{1/2}\right\|_F^2 + \alpha\left\|\alpha^{-1/4}\Sigma^{1/2}\boldsymbol{V}^\top\right\|_F^2\right).$$

This implies that there exists some $\boldsymbol{W} = \alpha^{1/4}\Sigma^{1/2}\boldsymbol{U}^\top$ and $\boldsymbol{H} = \alpha^{-1/4}\Sigma^{1/2}\boldsymbol{V}^\top$, such that $\|\boldsymbol{Z}\|_* = \frac{1}{2\sqrt{\alpha}}\left(\|\boldsymbol{W}\|_F^2 + \alpha\|\boldsymbol{H}\|_F^2\right)$, which further indicates that

$$\|\boldsymbol{Z}\|_* \geq \min_{\boldsymbol{Z} = \boldsymbol{W}^\top \boldsymbol{H}} \frac{1}{2\sqrt{\alpha}}\left(\|\boldsymbol{W}\|_F^2 + \alpha\|\boldsymbol{H}\|_F^2\right). \tag{19}$$

On the other hand, for any $\boldsymbol{W}^\top \boldsymbol{H} = \boldsymbol{Z}$, we have

$$\|\boldsymbol{Z}\|_* = trace(\Sigma) = trace(\boldsymbol{U}^\top \boldsymbol{Z}\boldsymbol{V}) = trace(\boldsymbol{U}^\top \boldsymbol{W}^\top \boldsymbol{H}\boldsymbol{V})$$

$$\leq \frac{1}{2\sqrt{\alpha}}\left\|\boldsymbol{U}^\top \boldsymbol{W}^\top\right\|_F^2 + \frac{\sqrt{\alpha}}{2}\|\boldsymbol{H}\boldsymbol{V}\|_F^2 = \frac{1}{2\sqrt{\alpha}}\left(\|\boldsymbol{W}\|_F^2 + \alpha\|\boldsymbol{H}\|_F^2\right),$$

where the first inequality is guaranteed by Young's inequality in Lemma C.2, and equality only holds when $\boldsymbol{W}\boldsymbol{U} = \sqrt{\alpha}\boldsymbol{H}\boldsymbol{V}$. The last equality follows because $\boldsymbol{U}\boldsymbol{U}^\top = \boldsymbol{I}$ and $\boldsymbol{V}\boldsymbol{V}^\top = \boldsymbol{I}$. Therefore, we have

$$\|\boldsymbol{Z}\|_* \leq \min_{\boldsymbol{Z} = \boldsymbol{W}^\top \boldsymbol{H}} \frac{1}{2\sqrt{\alpha}}\left(\|\boldsymbol{W}\|_F^2 + \alpha\|\boldsymbol{H}\|_F^2\right). \tag{20}$$

Combining the results in (19) and (20), we complete the proof.

$\square$

**Proposition C.4.** *Consider matrices $\boldsymbol{H} = [\boldsymbol{H}_1, \cdots, \boldsymbol{H}_n] \in \mathbb{R}^{d \times N}$ and $\boldsymbol{Z} = [\boldsymbol{Z}_1, \cdots, \boldsymbol{Z}_n] \in \mathbb{R}^{K \times N}$, where $\boldsymbol{H}_i \in \mathbb{R}^{d \times K}$ and $\boldsymbol{Z}_i \in \mathbb{R}^{K \times K}$ with $N = nK$. Let $\overline{\boldsymbol{H}} = \frac{1}{n}\sum_i \boldsymbol{H}_i$ and $\overline{\boldsymbol{Z}} = \frac{1}{n}\sum_i \boldsymbol{Z}_i$. Then $\boldsymbol{Z} = \boldsymbol{W}^\top \boldsymbol{H}$ indicates $\overline{\boldsymbol{Z}} = \boldsymbol{W}^\top \overline{\boldsymbol{H}}$. If $\overline{\boldsymbol{Z}}$ is a symmetric matrix, then we have*

$$\min_{\boldsymbol{Z} = \boldsymbol{W}^\top \boldsymbol{H}} \frac{1}{2\sqrt{\alpha}}\left(\|\boldsymbol{W}\|_F^2 + \alpha\|\boldsymbol{H}\|_F^2\right) = \min_{\overline{\boldsymbol{Z}} = \boldsymbol{W}^\top \overline{\boldsymbol{H}}} \frac{1}{2\sqrt{\alpha}}\left(\|\boldsymbol{W}\|_F^2 + \alpha n\|\overline{\boldsymbol{H}}\|_F^2\right) = \|\overline{\boldsymbol{Z}}\|_*, \tag{21}$$

*with the minimum is reached only if $\boldsymbol{H}_i = \overline{\boldsymbol{H}}$ (for $\forall i = 1, \cdots, n$) and $\boldsymbol{W} = \sqrt{\alpha n}\overline{\boldsymbol{H}}$.*

*Proof.* From Lemma C.2, we have $\|\boldsymbol{H}\|_F^2 = \sum_i \|\boldsymbol{H}_i\|_F^2 \geq n\|\overline{\boldsymbol{H}}\|_F^2$, with the equality hold only when $\boldsymbol{H}_i = \boldsymbol{H}_j$ for any $i \neq j$ . Consequently, this yields the following result:

$$\min_{\boldsymbol{Z}=\boldsymbol{W}^\top\boldsymbol{H}} \frac{1}{2\sqrt{\alpha}} \left(\|\boldsymbol{W}\|_F^2 + \alpha\|\boldsymbol{H}\|_F^2\right) = \min_{\overline{\boldsymbol{Z}}=\boldsymbol{W}^\top\overline{\boldsymbol{H}}} \frac{1}{2\sqrt{\alpha}} \left(\|\boldsymbol{W}\|_F^2 + \alpha n\|\overline{\boldsymbol{H}}\|_F^2\right).$$

Utilizing Lemma C.3, we further deduce:

$$\min_{\overline{\boldsymbol{Z}}=\boldsymbol{W}^\top\overline{\boldsymbol{H}}} \frac{1}{2\sqrt{\alpha}} \left(\|\boldsymbol{W}\|_F^2 + \alpha n\|\overline{\boldsymbol{H}}\|_F^2\right) = \|\overline{\boldsymbol{Z}}\|_*.$$

This minimum is achieved only when $\boldsymbol{W} = \sqrt{\alpha n}\overline{\boldsymbol{H}}$, thus completing the proof. □

# D   Proof of Theorem 4.1

## D.1   Proof of Theorem 4.1

In this section, we present the proof of Theorem 4.1 in Section 4, which we restate as follows.

**Theorem D.1.** *(Global Optimizer). Assume that the feature dimension d is no less than the number of classes K, i.e., $d \geq K$, and the dataset is balanced. Then any global optimizer of $(\boldsymbol{W}, \boldsymbol{H}, \boldsymbol{b})$ of*

$$\min_{\boldsymbol{W},\boldsymbol{H},\boldsymbol{b}} \mathcal{L}(\boldsymbol{W},\boldsymbol{H},\boldsymbol{b}) := \frac{1}{N} l_{CE}(\boldsymbol{W}^\top\boldsymbol{H} + \boldsymbol{b}\boldsymbol{1}_N^\top, \boldsymbol{Y}^\delta) + \frac{\lambda_W}{2}\|\boldsymbol{W}\|_F^2 + \frac{\lambda_H}{2}\|\boldsymbol{H}\|_F^2 + \frac{\lambda_b}{2}\|\boldsymbol{b}\|^2 \tag{22}$$

*satisfies the following properties:*

*The classification weight matrix $\boldsymbol{W}$ is given by*

$$\boldsymbol{W} = \left(\frac{\lambda_H n}{\lambda_W}\right)^{1/4} \sqrt{a^\delta} \boldsymbol{P} \left(\boldsymbol{I}_K - \boldsymbol{1}_K\boldsymbol{1}_K^\top/K\right). \tag{23}$$

*The matrix of last-layer feature $\boldsymbol{H}$ can be represented as*

$$\boldsymbol{H} = \overline{\boldsymbol{H}}\boldsymbol{Y}, \quad \overline{\boldsymbol{H}} = \left(\frac{\lambda_W}{\lambda_H n}\right)^{1/4} \sqrt{a^\delta} \boldsymbol{P} \left(\boldsymbol{I}_K - \boldsymbol{1}_K\boldsymbol{1}_K^\top/K\right). \tag{24}$$

*The bias $\boldsymbol{b}$ is a zero vector, i.e. $\boldsymbol{b} = \boldsymbol{0}$.*

*Here, $\boldsymbol{P} \in \mathbb{R}^{d \times K}(d \geq K)$ is a partial orthogonal matrix such that $\boldsymbol{P}^\top\boldsymbol{P} = \boldsymbol{I}_K$ and $a^\delta$ satisfies:*

(i) *if $\sqrt{KN}\lambda_Z + \delta \geq 1$, then $a^\delta = 0$;*

(ii) *if $\sqrt{KN}\lambda_Z + \delta < 1$, then*

$$a^\delta = \log\left(\frac{K}{\sqrt{KN}\lambda_Z + \delta} - K + 1\right).$$

*with $\lambda_Z = \sqrt{\lambda_W\lambda_H}$.*

*Proof.* The main idea of proving Theorem 4.1 is first to connect the problem (22) to its corresponding convex counterpart. This allows us to derive the precise form of the global minimizer for the convex optimization problem. Subsequently, we can further characterize the specific structures of $\boldsymbol{W}$ and $\boldsymbol{H}$ based on the acquired global minimizer.

**Connection of (22) to a Convex Problem.** Let $\boldsymbol{Z} = \boldsymbol{W}^\top\boldsymbol{H} \in \mathbb{R}^{K \times N}$ represent the output logit matrix with $N = nK$ and $\alpha = \frac{\lambda_H}{\lambda_W}$. Utilizing Lemma C.3, we get:

$$\min_{\boldsymbol{W}^\top\boldsymbol{H}=\boldsymbol{Z}} \lambda_W\|\boldsymbol{W}\|_F^2 + \lambda_H\|\boldsymbol{H}\|_F^2 = \sqrt{\lambda_W\lambda_H} \min_{\boldsymbol{W}^\top\boldsymbol{H}=\boldsymbol{Z}} \frac{1}{\sqrt{\alpha}} \left(\|\boldsymbol{W}\|_F^2 + \alpha\|\boldsymbol{H}\|_F^2\right) \tag{25}$$

$$= 2\sqrt{\lambda_W\lambda_H}\|\boldsymbol{Z}\|_*,$$

where $\|\boldsymbol{Z}\|_*$ represent nuclear norm of $\boldsymbol{Z}$. Additionally from Lemma C.3, the minimum is attained only when $\boldsymbol{H} = \overline{\boldsymbol{H}}\boldsymbol{Y}$ and $\boldsymbol{W} = \sqrt{\alpha n}\overline{\boldsymbol{H}} = \sqrt{\frac{\lambda_H}{\lambda_W n}}\overline{\boldsymbol{H}}$.

Let $\lambda_Z := \sqrt{\lambda_W \lambda_H}$, then Equation (22) becomes:

$$\min_{\boldsymbol{Z},\boldsymbol{b}} \mathcal{L}(\boldsymbol{Z},\boldsymbol{b}) := \frac{1}{N} l_{CE}(\boldsymbol{Z} + \boldsymbol{b}\mathbf{1}_N^\top, \boldsymbol{Y}^\delta) + \lambda_Z \|\boldsymbol{Z}\|_* + \frac{\lambda_b}{2}\|\boldsymbol{b}\|^2, \tag{26}$$

which is a convex optimization problem.

**Characterizing the Optimal Solution of (22) based on the Convex Program (26)**: We first derive the exact form of the global minimizer for the convex optimization problem (26). Particularly, we first establish that the predicted logit vectors within each class collapse to their sample means, i.e., $\boldsymbol{Z} = \overline{\boldsymbol{Z}}\boldsymbol{Y}$ (Lemma D.3). Subsequently, we derive the closed-form solution of $\overline{\boldsymbol{Z}}$ in Lemma D.5.

Furthermore, from Lemma C.3, we establish that the minimum in (25) is attained only if $\boldsymbol{H} = \overline{\boldsymbol{H}}\boldsymbol{Y}$ and $\boldsymbol{W} = \sqrt{\frac{\lambda_H}{\lambda_W n}}\overline{\boldsymbol{H}}$. Since $\boldsymbol{Z} = \boldsymbol{W}^\top \boldsymbol{H}$, combining the above result with Lemma D.5 yields the global minimizer $(\boldsymbol{W}, \boldsymbol{H}, \boldsymbol{b})$ of (22), satisfying:

$$\boldsymbol{W} = \left(\frac{\lambda_H n}{\lambda_W}\right)^{1/4} \sqrt{a^\delta} \boldsymbol{P}\left(\boldsymbol{I}_K - \mathbf{1}_K\mathbf{1}_K^\top/K\right), \tag{27}$$

$$\boldsymbol{H} = \overline{\boldsymbol{H}}\boldsymbol{Y}, \quad \overline{\boldsymbol{H}} = \left(\frac{\lambda_W}{\lambda_H n}\right)^{1/4} \sqrt{a^\delta} \boldsymbol{P}\left(\boldsymbol{I}_K - \mathbf{1}_K\mathbf{1}_K^\top/K\right), \tag{28}$$

and

$$\boldsymbol{b} = \mathbf{0},$$

where $\boldsymbol{P} \in \mathbb{R}^{d \times K}(d \geq K)$ is a partial orthogonal matrix such that $\boldsymbol{P}^\top \boldsymbol{P} = \boldsymbol{I}_K$ and $a^\delta$ is defined per the specifications in Lemma D.5.

Hence, we conclude the proof.

$\square$

## D.2 Supporting Lemma

**Lemma D.2.** *(Optimality Condition) The first-order optimality condition of $\mathcal{L}(\boldsymbol{Z},\boldsymbol{b})$ in Equation (26) is*

$$N^{-1}\left(\boldsymbol{Y}^\delta - \boldsymbol{P}\right) \in \lambda_Z \partial\|\boldsymbol{Z}\|_*, \quad N^{-1}\left(\boldsymbol{Y}^\delta - \boldsymbol{P}\right)\mathbf{1}_N = \lambda_b \boldsymbol{b}, \tag{29}$$

*where $\boldsymbol{P}$ is the prediction matrix defined as*

$$\boldsymbol{P} = \{\boldsymbol{p}_{ki}\}_{1 \leq k \leq K, 1 \leq i \leq n} \in \mathbb{R}^{K \times N}, \quad \boldsymbol{p}_{ki} := \frac{\exp(\boldsymbol{z}_{ki} + \boldsymbol{b})}{\langle\exp(\boldsymbol{z}_{ki} + \boldsymbol{b}), \mathbf{1}_K\rangle}, \tag{30}$$

*$\boldsymbol{Y}^\delta$ is the matrix of the smoothed soft target defined as*

$$\boldsymbol{Y}^\delta = (1 - \delta)\boldsymbol{Y} + \frac{\delta}{K}\mathbf{1}_K\mathbf{1}_N^\top, \quad \boldsymbol{Y} = [\boldsymbol{e}_1\mathbf{1}_n^\top, \cdots, \boldsymbol{e}_K\mathbf{1}_n^\top] \in \mathbb{R}^{K \times N}, \tag{31}$$

*and $\partial\|\boldsymbol{Z}\|_*$ represents the subdifferential of the nuclear norm of $\boldsymbol{Z}$.*

*Proof.* Consider

$$\mathcal{L}(\boldsymbol{Z},\boldsymbol{b}) = \frac{1}{N} l_{CE}(\boldsymbol{Z} + \boldsymbol{b}\mathbf{1}_N^\top, \boldsymbol{Y}^\delta) + \lambda_Z\|\boldsymbol{Z}\|_* + \frac{\lambda_b}{2}\|\boldsymbol{b}\|^2$$

in (26). Define

$$\phi(\boldsymbol{Z},\boldsymbol{b}) = \frac{1}{N} l_{CE}(\boldsymbol{Z} + \boldsymbol{b}\mathbf{1}_N^\top, \boldsymbol{Y}^\delta) = \frac{1}{N}\sum_{k=1}^K \sum_{i=1}^n l_{CE}(\boldsymbol{z}_{ki} + \boldsymbol{b}, \boldsymbol{y}_k^\delta), \tag{32}$$

where $\boldsymbol{y}_k^\delta = (1-\delta)\boldsymbol{e}_k + \frac{\delta}{K}\mathbf{1}_K$ is the smoothed target. The gradient of $\phi$ is

$$\frac{\partial \phi}{\partial \boldsymbol{z}_{ki}} = \frac{1}{N}\left(\boldsymbol{p}_{ki} - \boldsymbol{y}_k^\delta\right), \quad \frac{\partial \phi}{\partial \boldsymbol{b}} = \frac{1}{N}\sum_{k,i}\left(\boldsymbol{p}_{ki} - \boldsymbol{y}_k^\delta\right),$$

whose matrix form is:

$$\frac{\partial \phi}{\partial \boldsymbol{Z}} = \frac{1}{N}\left(\boldsymbol{P} - \boldsymbol{Y}^\delta\right), \quad \frac{\partial \phi}{\partial \boldsymbol{b}} = \frac{1}{N}\left(\boldsymbol{P} - \boldsymbol{Y}^\delta\right)\mathbf{1}_N.$$

Hence the gradient (subgradient) of $\mathcal{L}$ is

$$\frac{\partial \mathcal{L}}{\partial \boldsymbol{Z}} = N^{-1}(\boldsymbol{P} - \boldsymbol{Y}^\delta) + \lambda_Z\partial\|\boldsymbol{Z}\|_*, \quad \frac{\partial \mathcal{L}}{\partial \boldsymbol{b}} = N^{-1}(\boldsymbol{P} - \boldsymbol{Y}^\delta)\mathbf{1}_N + \lambda_b\boldsymbol{b},$$

where $\partial\|\boldsymbol{Z}\|_*$ is the subdifferential of the nuclear norm at $\boldsymbol{Z}$. Thus, $(\boldsymbol{Z}, \boldsymbol{b})$ is a global minimizer of $\mathcal{L}$ if its gradient (subgradient) is equal to zero, i.e.

$$N^{-1}(\boldsymbol{Y}^\delta - \boldsymbol{P}) \in \lambda_Z\partial\|\boldsymbol{Z}\|_*, \quad N^{-1}(\boldsymbol{Y}^\delta - \boldsymbol{P})\mathbf{1}_N = \lambda_b\boldsymbol{b}.$$

$\square$

**Lemma D.3.** *Assume that the number of classes $K$ is less than the feature dimension $d$, i.e., $K \leq d$, and the dataset is balanced. Then the prediction vectors, formulated as $\boldsymbol{z}_{ki} = f_\Theta(\boldsymbol{x}_{ki}), (1 \leq k \leq K, 1 \leq i \leq n)$ within each class collapse to their sample means $\bar{\boldsymbol{z}}_k$:*

$$\boldsymbol{z}_{ki} = \bar{\boldsymbol{z}}_k, \quad 1 \leq i \leq n, \tag{33}$$

*In other words, the prediction matrix $\boldsymbol{Z}$ can be written as the following factorized form:*

$$\boldsymbol{Z} = \overline{\boldsymbol{Z}}\boldsymbol{Y} \in \mathbb{R}^{K \times N} \tag{34}$$

*where*

$$\overline{\boldsymbol{Z}} = [\bar{\boldsymbol{z}}_1, \dots, \bar{\boldsymbol{z}}_K], \quad \boldsymbol{Y} = [\boldsymbol{e}_1\mathbf{1}_n^\top, \cdots, \boldsymbol{e}_K\mathbf{1}_n^\top].$$

*Proof.* The proof follows from the convexity of the loss function. Recall that the loss function in (26) was given by

$$\mathcal{L}(\boldsymbol{Z}, \boldsymbol{b}) := \phi(\boldsymbol{Z}, \boldsymbol{b}) + \lambda_Z\|\boldsymbol{Z}\|_* + \frac{\lambda_b}{2}\|\boldsymbol{b}\|^2,$$

where $\phi(\boldsymbol{Z}, \boldsymbol{b})$ is the Cross-Entropy (CE) or Label Smoothing (LS) loss (depending on the value of the smoothing parameter $\delta$) as defined in (32). Recalling that $\{\boldsymbol{z}_{ik}\}_{i=1}^n$ belong to the same class and $\bar{\boldsymbol{z}}_k = n^{-1}\sum_{i=1}^n \boldsymbol{z}_{ki}$ is their respective sample mean. From Jensen's inequality, we have

$$\begin{aligned}
\phi(\boldsymbol{Z}, \boldsymbol{b}) &= \frac{1}{Kn}\sum_{k=1}^K\sum_{i=1}^n l_{CE}(\boldsymbol{z}_{ki} + b, \boldsymbol{y}_k^\delta) \\
&\geq \frac{1}{K}\sum_{k=1}^K l_{CE}\left(\frac{1}{n}\sum_{i=1}^n(\boldsymbol{z}_{ki} + b), \boldsymbol{y}_k^\delta\right) \\
&= \frac{1}{K}\sum_{k=1}^K l_{CE}(\bar{\boldsymbol{z}}_k + \boldsymbol{b}, \boldsymbol{y}_k^\delta)
\end{aligned}$$

where the inequality becomes equality only when $\boldsymbol{z}_{ki} = \bar{\boldsymbol{z}}_k$.

In the rest of the proof, we employ a permutation argument. Let $\boldsymbol{Z}_l = [\boldsymbol{z}_{1l}, \cdots, \boldsymbol{z}_{Kl}]$ for $1 \leq l \leq n$ and $\tilde{\boldsymbol{Z}} = [\boldsymbol{Z}_1, \cdots, \boldsymbol{Z}_n]$. Let $\Gamma_i$ represent a distinct permutation of $n$. Consider $\tilde{\boldsymbol{Z}}_{\Gamma_i} = \tilde{\boldsymbol{Z}}\Pi_{\Gamma_i}$, where $\Pi_{\Gamma_i}$ is a permutation matrix rearranging $\tilde{\boldsymbol{Z}}$ so that the elements $\boldsymbol{Z}_l$ are ordered according to $\Gamma_i$. Then

$$\|\tilde{\boldsymbol{Z}}_{\Gamma_i}\|_* = \|\tilde{\boldsymbol{Z}}\|_* = \|\boldsymbol{Z}\|_*.$$

Since $\|\cdot\|_*$ is a convex function, we deduce

$$\|\boldsymbol{Z}\|_* = \frac{1}{n!}\left(\sum_i \|\tilde{\boldsymbol{Z}}_{\Gamma_i}\|_*\right) \geq \left\|\frac{1}{n!}\sum_i \tilde{\boldsymbol{Z}}_{\Gamma_i}\right\|_*,$$

where the inequality becomes equality only when $\boldsymbol{Z}_l = \overline{\boldsymbol{Z}}(1 \leq l \leq n)$ or equivalently $\boldsymbol{z}_{ki} = \bar{\boldsymbol{z}}_k$. As a result, it holds that

$$\mathcal{L}(\boldsymbol{Z}, \boldsymbol{b}) = \phi(\boldsymbol{Z}, \boldsymbol{b}) + \lambda_Z \|\boldsymbol{Z}\|_* + \frac{\lambda_b}{2}\|\boldsymbol{b}\|^2$$

$$\geq \frac{1}{K}\sum_{k=1}^K l_{CE}(\bar{\boldsymbol{z}}_k + \boldsymbol{b}, \boldsymbol{y}_k^\delta) + \lambda_Z \|\overline{\boldsymbol{Z}}\|_* + \frac{\lambda_b}{2}\|\boldsymbol{b}\|^2$$

$$= \frac{1}{N} l_{CE}\left(\overline{\boldsymbol{Z}}Y + \boldsymbol{b}\mathbf{1}_N^\top, \boldsymbol{Y}^\delta\right) + \lambda_Z \|\overline{\boldsymbol{Z}}\|_* + \frac{\lambda_b}{2}\|\boldsymbol{b}\|^2.$$

The global optimality of $\mathcal{L}(\boldsymbol{Z}, \boldsymbol{b})$ implies that $\boldsymbol{Z} = \overline{\boldsymbol{Z}}Y$, i.e., $\boldsymbol{z}_{ki} = \bar{\boldsymbol{z}}_k$ for $1 \leq k \leq n$.

$\square$

**Lemma D.4.** *Assume $\boldsymbol{z}_{ki} = \bar{\boldsymbol{z}}_k$, for $1 \leq k \leq K, 1 \leq i \leq n$ and $\langle \bar{\boldsymbol{z}}_k, \mathbf{1}_K \rangle = 0$, then it holds that*

$$N^{-1}\left((1-\delta)\boldsymbol{I}_K + \frac{\delta}{K}\boldsymbol{J}_K - \overline{\boldsymbol{P}}\right) = \lambda_Z\left(n^{-1/2}\left[\left(\overline{\boldsymbol{Z}}\overline{\boldsymbol{Z}}^\top\right)^\dagger\right]^{1/2}\overline{\boldsymbol{Z}} + \overline{\boldsymbol{R}}\right),$$

$$\frac{n}{N}\left((1-\delta)\boldsymbol{I}_K + \frac{\delta}{K}\boldsymbol{J}_K - \overline{\boldsymbol{P}}\right)\mathbf{1}_K = \lambda_b\boldsymbol{b},$$

*where $\boldsymbol{J}_K \in \mathbb{R}^{K\times K}$ is the ones matrix, $\overline{\boldsymbol{Z}} = [\bar{\boldsymbol{z}}_1, \cdots, \bar{\boldsymbol{z}}_K] \in \mathbb{R}^{K\times K}$, and $\overline{\boldsymbol{R}}$ satisfies $\overline{\boldsymbol{R}}\overline{\boldsymbol{Z}}^\top = 0$, $\overline{\boldsymbol{Z}}^\top\overline{\boldsymbol{R}} = 0$, and $\|\overline{\boldsymbol{R}}\| \leq n^{-1/2}$.*

*In particular, if $\overline{\boldsymbol{Z}}$ is of rank $K-1$, then*

$$N^{-1}\left((1-\delta)\boldsymbol{I}_K + \frac{\delta}{K}\boldsymbol{J}_K - \overline{\boldsymbol{P}}\right) = \frac{\lambda_Z}{\sqrt{n}}\left(\left[\left(\overline{\boldsymbol{Z}}\overline{\boldsymbol{Z}}^\top\right)^\dagger\right]^{1/2}\overline{\boldsymbol{Z}}\right).$$

*Proof.* Under the assumption, $\boldsymbol{z}_{ki} = \bar{\boldsymbol{z}}_k$, $1 \leq i \leq n$, and $\langle \bar{\boldsymbol{z}}_k, \mathbf{1}_K \rangle = 0$, we have $\boldsymbol{Z} = \overline{\boldsymbol{Z}}Y$ and $\boldsymbol{P} = \overline{\boldsymbol{P}}Y$, where $\overline{\boldsymbol{P}}$ is defined as

$$\overline{\boldsymbol{P}} = [\bar{\boldsymbol{p}}_1, \cdots, \bar{\boldsymbol{p}}_K]$$

with $\bar{\boldsymbol{p}}_k$ as the probability vector w.r.t. the $\bar{\boldsymbol{z}}_k + \boldsymbol{b}$. Then the optimality condition in (29) reduces to

$$N^{-1}\left((1-\delta)\boldsymbol{I}_K + \frac{\delta}{K}\boldsymbol{J}_K - \overline{\boldsymbol{P}}\right)Y \in \lambda_Z\partial\|\boldsymbol{Z}\|_*. \tag{35}$$

Let $\boldsymbol{Z} = \boldsymbol{U}\Sigma\boldsymbol{V}^\top$ be the SVD of $\boldsymbol{Z}$, then we have:

$$\partial\|\boldsymbol{Z}\|_* = \{\boldsymbol{U}\boldsymbol{V}^\top + \boldsymbol{R} : \|\boldsymbol{R}\| \leq 1, \boldsymbol{U}^\top\boldsymbol{R} = 0, \boldsymbol{R}\boldsymbol{V} = 0\},$$

where

$$\boldsymbol{U}\boldsymbol{V}^\top = \left[\left(\boldsymbol{Z}\boldsymbol{Z}^\top\right)^\dagger\right]^{1/2}\boldsymbol{Z}.$$

Since $\boldsymbol{Z} = \overline{\boldsymbol{Z}}Y$ and $\boldsymbol{Y}\boldsymbol{Y}^\top = n\boldsymbol{I}_K$, we further get:

$$\boldsymbol{U}\boldsymbol{V}^\top = n^{-1/2}\left[\left(\overline{\boldsymbol{Z}}\overline{\boldsymbol{Z}}^\top\right)^\dagger\right]^{1/2}\overline{\boldsymbol{Z}}Y.$$

Then (35) is equivalent to:

$$N^{-1}\left((1-\delta)\boldsymbol{I}_K + \frac{\delta}{K}\boldsymbol{J}_K - \overline{\boldsymbol{P}}\right)Y = \lambda_Z\left(n^{-1/2}\left[\left(\overline{\boldsymbol{Z}}\overline{\boldsymbol{Z}}^\top\right)^\dagger\right]^{1/2}\overline{\boldsymbol{Z}}Y + \boldsymbol{R}\right),$$

where $\boldsymbol{R}$ is in the form of

$$\boldsymbol{R} = \overline{\boldsymbol{R}}\boldsymbol{Y}, \quad \overline{\boldsymbol{R}} = [\bar{\boldsymbol{r}}_1, \cdots, \bar{\boldsymbol{r}}_K]$$

such that

$$\overline{\boldsymbol{R}}\boldsymbol{Y}\left(\overline{\boldsymbol{Z}}\boldsymbol{Y}\right)^\top = n\overline{\boldsymbol{R}}\overline{\boldsymbol{Z}}^\top = 0, \quad \overline{\boldsymbol{Z}}^\top \overline{\boldsymbol{R}} = 0, \quad \|n^{1/2}\overline{\boldsymbol{R}}\| \leq 1.$$

This further leads to

$$N^{-1}\left((1-\delta)\boldsymbol{I}_K + \frac{\delta}{K}\boldsymbol{J}_K - \overline{\boldsymbol{P}}\right) = \lambda_Z\left(n^{-1/2}\left[(\overline{\boldsymbol{Z}}\overline{\boldsymbol{Z}}^\top)^\dagger\right]^{1/2}\overline{\boldsymbol{Z}} + \overline{\boldsymbol{R}}\right)$$

where $\overline{\boldsymbol{R}}\overline{\boldsymbol{Z}}^\top = 0$, $\overline{\boldsymbol{Z}}^\top \overline{\boldsymbol{R}} = 0$ and $\|\overline{\boldsymbol{R}}\| \leq n^{-1/2}$.

For $\boldsymbol{b}$, it is easy to see that the optimality condition in equation 29 reduces to

$$\lambda_b \boldsymbol{b} = N^{-1}\left(\boldsymbol{Y}^\delta - \boldsymbol{P}\right)\mathbf{1}_N = \frac{n}{N}\left((1-\delta)\boldsymbol{I}_K + \frac{\delta}{K}\boldsymbol{J}_K - \overline{\boldsymbol{P}}\right)\mathbf{1}_K.$$

Now, if $\overline{\boldsymbol{Z}}$ is of rank $K-1$, since $\mathbf{1}_K^\top \overline{\boldsymbol{Z}} = 0$, the columns of $\overline{\boldsymbol{R}}$ are parallel to $\mathbf{1}_K$. Moreover, $\overline{\boldsymbol{P}}$ is a positive left stochastic matrix with $\overline{\boldsymbol{P}}^\top \mathbf{1}_K = \mathbf{1}_K$, and therefore $\boldsymbol{I}_K - \overline{\boldsymbol{P}}$ is also of rank $K-1$. The left stochasticity of $\overline{\boldsymbol{P}}$ further deduces that $(\boldsymbol{I}_K - \overline{\boldsymbol{P}})^T \mathbf{1}_K = 0$. In other words, $\mathbf{1}_K$ is in the left null space of $\boldsymbol{I}_K - \overline{\boldsymbol{P}}$, which in turn implies $\overline{\boldsymbol{R}} = 0$. This leads to

$$N^{-1}\left((1-\delta)\boldsymbol{I}_K + \frac{\delta}{K}\boldsymbol{J}_K - \overline{\boldsymbol{P}}\right) = \frac{\lambda_Z}{\sqrt{n}}\left(\left[(\overline{\boldsymbol{Z}}\overline{\boldsymbol{Z}}^\top)^\dagger\right]^{1/2}\overline{\boldsymbol{Z}}\right).$$

$\square$

**Lemma D.5.** *Assume that the number of classes $K$ is less than or equal to the feature dimension $d$, i.e., $K \leq d$, and the dataset is balanced. Then the global minimizer $(\boldsymbol{Z}, \boldsymbol{b})$ of (26) satisfies the following properties:*

$$\boldsymbol{Z} = \overline{\boldsymbol{Z}}\boldsymbol{Y}, \quad \overline{\boldsymbol{Z}} = a^\delta(\boldsymbol{I}_K - \boldsymbol{J}_K/K), \quad \boldsymbol{b} = 0. \tag{36}$$

*In Particular, we have*

*(i) if $\sqrt{KN}\lambda_Z + \delta \geq 1$, then $a^\delta = 0$;*

*(ii) if $\sqrt{KN}\lambda_Z + \delta < 1$, then*

$$a^\delta = \log\left(\frac{K}{\sqrt{KN}\lambda_Z + \delta} - K + 1\right)$$

*Proof.* From Lemma D.3, we have $\boldsymbol{Z} = \overline{\boldsymbol{Z}}\boldsymbol{Y}$. Moreover, according to Lemma 8 in (Zhou et al., 2022b), there exists a constant $a$ such that the global optimum of (26) satisfies:

$$\boldsymbol{Z} = a\left(\boldsymbol{I}_K - \mathbf{1}_K \mathbf{1}_K^\top/K\right)\boldsymbol{Y}, \quad \boldsymbol{b} = 0.$$

Equivalently, we have $\overline{\boldsymbol{Z}} = a\left(\boldsymbol{I}_K - \mathbf{1}_K \mathbf{1}_K^\top/K\right)$. Further, we define $\overline{\boldsymbol{P}} = [\bar{\boldsymbol{p}}_1, \cdots, \bar{\boldsymbol{p}}_K]$ with $\bar{\boldsymbol{p}}_k$ as the probability vector w.r.t. the logit $\boldsymbol{z}_k + \boldsymbol{b}$. Then,

$$\overline{\boldsymbol{P}} = \frac{\boldsymbol{J}_K + (e^a - 1)\boldsymbol{I}_K}{K - 1 + e^a}, \tag{37}$$

and

$$(1-\delta)\boldsymbol{I}_K + \frac{\delta}{K}\boldsymbol{J}_K - \overline{\boldsymbol{P}} = (1-\delta)\boldsymbol{I}_K + \frac{\delta}{K}\boldsymbol{J}_K - \frac{\boldsymbol{J}_K + (e^a - 1)\boldsymbol{I}_K}{K - 1 + e^a}$$

$$= \left(\frac{K}{K - 1 + e^a} - \delta\right)\left(\boldsymbol{I}_K - \frac{1}{K}\boldsymbol{J}_K\right)$$

Recall that the first-order optimality condition as expressed in Lemma D.2 indicates that:

$$(1-\delta)\boldsymbol{I}_K + \frac{\delta}{K}\boldsymbol{J}_K - \overline{\boldsymbol{P}} = N\lambda_Z \left( n^{-1/2} \left[ \left( \overline{\boldsymbol{Z}}\overline{\boldsymbol{Z}}^\top \right)^\dagger \right]^{1/2} \overline{\boldsymbol{Z}} + \overline{\boldsymbol{R}} \right) \tag{38}$$

$$= N\lambda_Z \left( \text{sign}(a)\sqrt{\frac{1}{n}}(\boldsymbol{I}_K - \boldsymbol{J}_K/K) + \overline{\boldsymbol{R}} \right). \tag{39}$$

Suppose $a \neq 0$, then $\overline{\boldsymbol{Z}}$ has rank $K-1$ and we have $\overline{\boldsymbol{R}} = 0$. Thus the above optimality condition implies:

$$\left( \frac{K}{K-1+e^a} - \delta \right) (\boldsymbol{I}_K - \boldsymbol{J}_K/K) = \text{sign}(a)\sqrt{KN}\lambda_Z(\boldsymbol{I}_K - \boldsymbol{J}_K/K),$$

which is equivalent to

$$\frac{K}{K-1+e^a} - \delta = \text{sign}(a)\sqrt{NK}\lambda_Z.$$

For $a > 0$, the above equation has the following solution:

$$a = \log\left( \frac{K}{\sqrt{NK}\lambda_Z + \delta} - K + 1 \right) \quad \text{if} \quad \sqrt{NK}\lambda_Z + \delta < 1.$$

If $\sqrt{NK}\lambda_Z + \delta \geq 1$, then we select $a = 0$, and noting that in that case $\overline{\boldsymbol{P}} = K^{-1}\boldsymbol{J}_K$, we have that the optimality condition in (38) implies that $\overline{\boldsymbol{R}}$ satisfies

$$(1-\delta)\boldsymbol{I}_K + \frac{\delta-1}{K}\boldsymbol{J}_K = N\lambda_Z\overline{\boldsymbol{R}}.$$

We get

$$\overline{\boldsymbol{R}} = \frac{1}{N\lambda_Z}\left[ (1-\delta)\boldsymbol{I}_K + \frac{\delta-1}{K}\boldsymbol{J}_K \right],$$

where $\|\overline{\boldsymbol{R}}\| \leq n^{-1/2}$ meets the requirement of the optimality condition.

$\square$

# E   Proof of Theorem 4.2

In this section of the appendices, we prove Theorem 4.2.

*Proof.* To provide insight into the accelerated convergence of the model under label smoothing loss, we examine the Hessian matrix of the empirical loss function, defined as follows:

$$\min_{\boldsymbol{W},\boldsymbol{H}} \phi(\boldsymbol{W}, \boldsymbol{H}) := \frac{1}{N}l_{CE}(\boldsymbol{W}^\top\boldsymbol{H}, \boldsymbol{Y}^\delta). \tag{40}$$

Under commonly used deep learning frameworks, the model parameters are updated iteratively, and it is common and often practical to analyze the Hessian matrix with respect to $\boldsymbol{H}, \boldsymbol{W}$ individually rather than considering the full Hessian. Particularly, we demonstrate that the Hessian of the empirical loss concerning $\boldsymbol{W}$ when $\boldsymbol{H}$ is fixed and the Hessian w.r.t. $\boldsymbol{H}$ when $\boldsymbol{W}$ is fixed are positive semi-definite at the global optimizer under both cross-entropy and label smoothing losses. Furthermore, we establish that the condition numbers of these Hessian matrices are notably lower under the LS loss in comparison to the CE loss.

**Hessian matrix with respect to $\boldsymbol{Z}$.** Let $\boldsymbol{Z} = \boldsymbol{W}^\top\boldsymbol{H} \in \mathbb{R}^{K\times N}$ represent the prediction logit matrix. In the proof of Lemma D.2, we obtained the first-order partial derivatives of the loss $\phi$:

$$\frac{\partial\phi}{\partial\boldsymbol{z}_{ki}} = \frac{1}{N}\left( \boldsymbol{p}_{ki} - \boldsymbol{y}_k^\delta \right), \quad \frac{\partial\phi}{\partial\boldsymbol{b}} = \frac{1}{N}\sum_{k,i}\left( \boldsymbol{p}_{ki} - \boldsymbol{y}_k^\delta \right),$$

where $\boldsymbol{p}_{ki}$ is the prediction for the $i$-th sample that belongs to the $k$-th class and $\boldsymbol{y}_k^\delta$ is the soft label for class $k$ with a smoothing parameter $\delta$. These partial derivatives lead to the corresponding second-order partial derivatives of the loss $\phi$:

$$\frac{\partial^2 \phi}{\partial \boldsymbol{z}_{ki}^2} = \frac{1}{N} \left( \operatorname{diag}(\boldsymbol{p}_{ki}) - \boldsymbol{p}_{ki}\boldsymbol{p}_{ki}^\top \right), \quad \frac{\partial^2 \phi}{\partial \boldsymbol{z}_{ki} \partial \boldsymbol{z}_{k'i'}} = 0, \forall (k,i) \neq (k',i'),$$

$$\frac{\partial^2 \phi}{\partial \boldsymbol{b}^2} = \frac{1}{N} \sum_{k,i} \left( \operatorname{diag}(\boldsymbol{p}_{ki}) - \boldsymbol{p}_{ki}\boldsymbol{p}_{ki}^\top \right), \quad \frac{\partial^2 \phi}{\partial \boldsymbol{z}_{ki} \partial \boldsymbol{b}} = \frac{1}{N} \left( \operatorname{diag}(\boldsymbol{p}_{ki}) - \boldsymbol{p}_{ki}\boldsymbol{p}_{ki}^\top \right).$$

From Theorem 4.1, we have the global optimizer satisfies $\boldsymbol{z}_{ki}^2 = \bar{\boldsymbol{z}}_k$ and $\boldsymbol{p}_{ki}^2 = \bar{\boldsymbol{p}}_k, 1 \leq k \leq K, 1 \leq i \leq n$. Consequently, it follows that $\frac{\partial^2 \phi}{\partial \boldsymbol{z}_{ki}^2} = \frac{\partial^2 \phi}{\partial \boldsymbol{z}_{ki'}^2} (\forall 1 \leq i, i' \leq n)$. To simply the notation, we denote

$$\boldsymbol{D}_k = \operatorname{diag}(\bar{\boldsymbol{p}}_k) - \bar{\boldsymbol{p}}_k \bar{\boldsymbol{p}}_k^\top. \tag{41}$$

Then we have $\frac{\partial^2 \phi}{\partial \boldsymbol{z}_{ki}^2} = \boldsymbol{D}_k/N$.

The closed-form solution for $\overline{\boldsymbol{P}}$ in (37) yields the expression:

$$\bar{\boldsymbol{p}}_k = \frac{1}{K - 1 + e^{a^\delta}} \left( (e^{a^\delta} - 1)\boldsymbol{e}_k + \boldsymbol{1}_K \right). \tag{42}$$

To simplify the notation, we denote

$$p_t = e^{a^\delta}/(K - 1 + e^{a^\delta}), \quad p_n = 1/(K - 1 + e^{a^\delta}), \tag{43}$$

where $p_t$ ($p_n$) is the predicted probability for the target (non-target) class at the global optimal solution. Under this notation, we have $p_t + (K - 1)p_n = 1$ and $\bar{\boldsymbol{p}}_k = (p_t - p_n)\boldsymbol{e}_k + p_n \boldsymbol{1}_K$.

For any $1 \leq k \leq K$, $\boldsymbol{p}_k \boldsymbol{p}_k^\top$ is a positive matrix and the associated Laplacian $\boldsymbol{D}_k$ is positive-semidefinite with all eigenvalues non negative. The smallest eigenvalue of the Laplacian matrix $\boldsymbol{D}_k$ is $\sigma_1 = 0$ with the corresponding eigenvector $\boldsymbol{v}_1 = \boldsymbol{1}_K/\|\boldsymbol{1}_K\|$.

Define

$$\boldsymbol{v}_2 = (K\boldsymbol{e}_1 - \boldsymbol{1}_K)/\|K\boldsymbol{e}_1 - \boldsymbol{1}_K\|,$$

then we have

$$\begin{aligned} \boldsymbol{D}_k \boldsymbol{v}_2 &= \operatorname{diag}(\bar{\boldsymbol{p}}_k) \boldsymbol{v}_2 - \bar{\boldsymbol{p}}_k \bar{\boldsymbol{p}}_k^\top \boldsymbol{v}_2 \\ &= K p_t p_n \boldsymbol{v}_2, \end{aligned}$$

from which we get $K p_t p_n$ is an eigenvalue of $\boldsymbol{D}_k$ with the corresponding eigenvector $\boldsymbol{v}_2$.

Consider the outer-product matrix $\boldsymbol{p}_k \boldsymbol{p}_k^\top$, its largest eigenvalue is $\|\boldsymbol{p}_k\|^2$ with the eigenvector $\boldsymbol{p}_k/\|\boldsymbol{p}_k\|$. Its null space is of dimension $K - 1$. Particularly, we can find a set of the basis vectors for $\operatorname{Null}(\boldsymbol{p}_k \boldsymbol{p}_k^\top)$ as follows:

$$\{\boldsymbol{o}_1, \cdots, \boldsymbol{o}_{K-1} : \operatorname{span}(\boldsymbol{o}_1, \boldsymbol{p}_k) = \operatorname{span}(\boldsymbol{e}_k, \boldsymbol{p}_k)\}, \tag{44}$$

which means the vectors $\boldsymbol{o}_1$ and $\boldsymbol{p}_k$ span the same 2D space as $\boldsymbol{e}_k$ and $\boldsymbol{p}_k$.

Note that

$$\operatorname{span}(\boldsymbol{e}_k, \boldsymbol{p}_k) = \operatorname{span}(\boldsymbol{v}_1, \boldsymbol{v}_2),$$

where $\boldsymbol{v}_1, \boldsymbol{v}_2$ are the eigenvector for $\boldsymbol{D}_k$.

Then we can show that $\boldsymbol{o}_2 \cdots, \boldsymbol{o}_{K-1}$ are eigenvectors of $\boldsymbol{D}_k$ with the corresponding eigenvalue as $\sigma_3 = p_n$. Particularly, with $\boldsymbol{o}_l$ orthonormal to both $\bar{\boldsymbol{p}}_k$ and $\boldsymbol{e}_k$, we have

$$
\begin{aligned}
\boldsymbol{D}_k \boldsymbol{o}_l &= \mathrm{diag}\left(\bar{\boldsymbol{p}}_k\right) \boldsymbol{o}_l - \bar{\boldsymbol{p}}_k \bar{\boldsymbol{p}}_k^\top \boldsymbol{o}_l \\
&= \mathrm{diag}\left(\bar{\boldsymbol{p}}_k\right) \boldsymbol{o}_l \\
&= p_n \boldsymbol{o}_l,
\end{aligned}
$$

for $\boldsymbol{o}_l (2 \le l \le K - 1)$ defined in (44).

To summarize the unique eigenvalues of the matrix $\boldsymbol{D}$ are

$$
\sigma_1 = 0, \quad \sigma_2 = p_n, \quad \sigma_3 = K p_t p_n \tag{45}
$$

with $p_t$ and $p_n$ defined in (43).

**Hessian matrix with respect to $\boldsymbol{H}$.** The gradient of $\phi$ with respect to $\boldsymbol{h}_{ki}$ is

$$
\frac{\partial \phi}{\partial \boldsymbol{h}_{ki}} = \frac{\partial \boldsymbol{z}_{ki}}{\partial \boldsymbol{h}_{ki}} \frac{\partial \phi}{\partial \boldsymbol{z}_{ki}} = \boldsymbol{W} \frac{\partial \phi}{\partial \boldsymbol{z}_{ki}}.
$$

Further, we can easily get the corresponding Hessian:

$$
\frac{\partial^2 \phi}{\partial \boldsymbol{h}_{ki}^2} = \boldsymbol{W} \frac{\partial^2 \phi}{\partial \boldsymbol{z}_{ki}^2} \boldsymbol{W}^\top = \frac{1}{N} \boldsymbol{W} \boldsymbol{D}_k \boldsymbol{W}^\top, \quad \frac{\partial^2 \phi}{\partial \boldsymbol{h}_{ki} \partial \boldsymbol{h}_{k'i'}} = 0, \tag{46}
$$

where $\boldsymbol{D}_k$ is the Laplacian matrix as defined in (41).

From Theorem 4.1, we have

$$
\boldsymbol{W} = \left(\frac{\lambda_H n}{\lambda_W}\right)^{1/4} \sqrt{a^\delta} \boldsymbol{P} \left(\boldsymbol{I}_K - \boldsymbol{J}_K / K\right).
$$

For simplicity, we denote $a_W^\delta = (\lambda_H n / \lambda_W)^{1/4} \sqrt{a^\delta}$, then $\boldsymbol{W} = a_W^\delta \boldsymbol{P} (\boldsymbol{I}_K - \boldsymbol{J}_K / K)$. Under this notation, we have

$$
\frac{1}{N} \boldsymbol{W} \boldsymbol{D}_k \boldsymbol{W}^\top = \frac{(a_W^\delta)^2}{N} \boldsymbol{P} (\boldsymbol{I}_K - \boldsymbol{J}_K / K) \boldsymbol{D}_k (\boldsymbol{I}_K - \boldsymbol{J}_K / K) \boldsymbol{P}^\top \tag{47}
$$

$$
= \frac{(a_W^\delta)^2}{N} \boldsymbol{P} \boldsymbol{D}_k \boldsymbol{P}^\top. \tag{48}
$$

Note that $\boldsymbol{P} \in \mathbb{R}^{d \times K}, (d > K)$ is a partial orthogonal matrix. Given the eigenvalues of $\boldsymbol{D}_k$ in (45), the eigenvalues for (47) are:

$$
\lambda_1 = 0, \quad \lambda_2 = \frac{(a_W^\delta)^2}{N} p_n, \quad \lambda_3 = \frac{(a_W^\delta)^2}{N} K p_t p_n \tag{49}
$$

with the corresponding multiplicities $m(\lambda_1) = 1 + d - K$, $m(\lambda_2) = K - 2$, $m(\lambda_3) = 1$.

From (52), the hessian of $\phi$ with respect to $\boldsymbol{h} = \mathrm{vec}(\boldsymbol{H})$ is a block diagonal matrix which can be expressed as

$$
\frac{\partial^2 \phi}{\partial \boldsymbol{h}^2} = \frac{1}{N} \mathrm{blkdiag}(\boldsymbol{W} \boldsymbol{D}_1 \boldsymbol{W}^\top, \cdots, \boldsymbol{W} \boldsymbol{D}_1 \boldsymbol{W}^\top, \cdots, \boldsymbol{W} \boldsymbol{D}_K \boldsymbol{W}^\top, \cdots, \boldsymbol{W} \boldsymbol{D}_K \boldsymbol{W}^\top). \tag{50}
$$

The unique eigenvalues of the Hessian matrix $\frac{\partial^2 \phi}{\partial \boldsymbol{h}^2}$ are the same as provided in (49). Given that the Hessian matrix contains a zero eigenvalue, our analysis centers on its condition number within the non-zero eigenvalue space. This is calculated as follows:

$$
\kappa(\nabla_{\boldsymbol{h}}^2 \phi) = \lambda_3 / \lambda_2 = K p_t \tag{51}
$$

Considering the formula for $p_t$ given in (43), an increase in $\delta$ leads to a decrease in $p_t$, subsequently resulting in a smaller condition number.

**Hessian matrix with respect to $W$**. The gradient of $\phi$ with respect to $w_l(l = 1, \cdots, K)$ is

$$\frac{\partial \phi}{\partial w_l} = \frac{\partial z_{ki}}{\partial w_l} \frac{\partial \phi}{\partial z_{ki}}.$$

Further, we get the corresponding Hessian:

$$\frac{\partial^2 \phi}{\partial w_l \partial w_{l'}} = \sum_{k,i} \frac{\partial z_{ki}}{\partial w_{l'}} \frac{\partial^2 \phi}{\partial z_{ki}^2} \left( \frac{\partial z_{ki}}{\partial w_l} \right)^\top$$

$$= \sum_{k,i} h_{ki} e_{l'}^\top \frac{\partial^2 \phi}{\partial z_{ki}^2} e_l h_{ki}^\top$$

$$= \sum_{k,i} D_k(l', l) h_{ki} h_{ki}^\top$$

$$= n \sum_k D_k(l', l) \bar{h}_k \bar{h}_k^\top$$

where $D_k(l', l)$ is $(l', l)$-th element in $D_k$ and $\bar{h}_k$ is the $k$-th column vector in $\overline{H}$ as defined in 24. From the defination of $D_k$ in (41), we have

$$\frac{\partial^2 \phi}{\partial w_l \partial w_{l'}} = \begin{cases} n\overline{H} \left[ \mathrm{diag}(\bar{p}_l) - \mathrm{diag}(\bar{p}_l)^2 \right] \overline{H}^\top & \text{if } l = l' \\ n\overline{H} \left[ -\mathrm{diag}(\bar{p}_l)\mathrm{diag}(\bar{p}_{l'}) \right] \overline{H}^\top & \text{if } l \neq l' \end{cases}$$

where $\bar{p}_l$ is the average prediction vector as defined in (42). Particularly, the $l$-th element of $\bar{p}_l$ equals $p_t$ and the others equal $p_n$.

From (24), we have

$$\overline{H} = \left( \frac{\lambda_W}{\lambda_H n} \right)^{1/4} \sqrt{a^\delta} P \left( I_K - J_K/K \right).$$

Hence, the Hessian of $\phi$ w.r.t. the $w = \mathrm{vec}(W)$ can be written as:

$$\frac{\partial^2 \phi}{\partial w^2} = n \left( \frac{\lambda_W}{\lambda_H n} \right)^{1/2} a^\delta D_P D_\Pi S D_\Pi D_P^\top \tag{52}$$

where

$$D_P = \mathrm{blkdiag}(P, \cdots, P) \in \mathbb{R}^{Kd \times K^2}, \tag{53}$$

$$D_\Pi = \mathrm{blkdiag}(\Pi, \cdots, \Pi) \in \mathbb{R}^{K^2 \times K^2}, \quad \Pi = I_K - \frac{1_K 1_K^\top}{K} \tag{54}$$

and

$$S = \begin{bmatrix} \Lambda_1 & & \\ & \ddots & \\ & & \Lambda_K \end{bmatrix} - \begin{bmatrix} \Lambda_1 \\ \vdots \\ \Lambda_K \end{bmatrix} \begin{bmatrix} \Lambda_1 & \cdots & \Lambda_K \end{bmatrix} \tag{55}$$

with $\Lambda_l = \mathrm{diag}(\bar{p}_l)$.

Since $P$ is a partial orthogonal matrix, the condition number of (52) is the same as the condition number of the following matrix:

$$B = D_\Pi S D_\Pi. \tag{56}$$

Subsequently, we proceed to derive the eigenvalues for the above matrix $B$.

First, considering an eigenvector $v$ of $B$, let $\Delta$ be a $K \times K$ matrix such that $v = \mathrm{vec}(\Delta)$. Considering any $\Delta$ satisfying $\Delta\Pi = 0$ or $\Delta^\top \Pi = 0$, for the corresponding $v = \mathrm{vec}(\Delta)$ we get $D_\Pi v = 0$ and $v^\top D_\Pi = 0$, wich

further yields $\boldsymbol{B}\boldsymbol{v} = 0$. Since $\text{rank}(\boldsymbol{\Pi}) = K - 1$, it follows that $\lambda_1 = 0$ is an eigenvalue of $\boldsymbol{B}$ with multiplicity $2K - 1$.

On the other hand, consider any $\Delta$ that satisfies the conditions:

$$\text{diag}(\Delta) = 0, \quad \Delta \boldsymbol{1}_K = \Delta^\top \boldsymbol{1}_K = 0. \tag{57}$$

It is noteworthy that from $\Delta \boldsymbol{1}_K = \Delta^\top \boldsymbol{1}_K = 0$, the corresponding vector $\boldsymbol{v}$ satisfies $\boldsymbol{D}_\Pi \boldsymbol{v} = \boldsymbol{v}$. If $\Delta$ further satisfies $\text{diag}(\Delta) = 0$ then $\boldsymbol{S}\boldsymbol{v} = p_n \boldsymbol{v}$. Consequently, we deduce that

$$\boldsymbol{B}\boldsymbol{v} = \boldsymbol{D}_\Pi \boldsymbol{S} \boldsymbol{D}_\Pi \boldsymbol{v} = \boldsymbol{D}_\Pi \boldsymbol{S} \boldsymbol{v} = \boldsymbol{D}_\Pi p_n \boldsymbol{v} = p_n \boldsymbol{v}$$

We can conclude that $p_n$ is an eigenvalue of $\boldsymbol{B}$ with multiplicity $K^2 - K - (2K - 1) = K^2 - 3K + 1$.

Next, let us consider the matrix $\Delta = \boldsymbol{\Pi}\text{diag}(\boldsymbol{u})\boldsymbol{\Pi}$, where $\boldsymbol{u}$ is a vector satisfying $\boldsymbol{u}^\top \boldsymbol{1}_K = 0$. The vectorized form of $\Delta$ can be denoted as

$$\boldsymbol{v} = \text{vec}(\Delta) = \begin{bmatrix} \boldsymbol{\Pi}^\top \text{diag}(\boldsymbol{u})\boldsymbol{\Pi}_1 \\ \boldsymbol{\Pi}^\top \text{diag}(\boldsymbol{u})\boldsymbol{\Pi}_2 \\ \vdots \\ \boldsymbol{\Pi}^\top \text{diag}(\boldsymbol{u})\boldsymbol{\Pi}_K \end{bmatrix},$$

where $\boldsymbol{\Pi}_k$ represents the $k$-th column of the matrix $\boldsymbol{\Pi}$. Due to the properties $\boldsymbol{\Pi} = \boldsymbol{\Pi}^\top$ and $\boldsymbol{\Pi}^2 = \boldsymbol{\Pi}$, it follows that $\boldsymbol{D}_\Pi \boldsymbol{v} = \boldsymbol{v}$. Additionally, given $\boldsymbol{u}^\top \boldsymbol{1}_K = 0$, we obtain $\boldsymbol{S}\boldsymbol{v} = \lambda_2 \boldsymbol{v}$ with $\lambda_2 = (1 - p_t + p_n) \cdot \frac{p_n + (K-1)p_t}{K}$. Consequently, we conclude that $\lambda_2 = (1 - p_t + p_n) \cdot \frac{p_n + (K-1)p_t}{K}$ is an eigenvalue of $\boldsymbol{B}$ with a multiplicity equal to the degrees of freedom of $\boldsymbol{u}$, which is $K - 1$.

Lastly, considering $\boldsymbol{v} = \text{vec}(\boldsymbol{\Pi})$, we observe that $\boldsymbol{D}_\Pi \boldsymbol{v} = \boldsymbol{v}$. Further, we have $\boldsymbol{S}\boldsymbol{v} = Kp_t p_n \boldsymbol{v}$. Hence, we obtain

$$\boldsymbol{B}\boldsymbol{v} = \boldsymbol{D}_\Pi \boldsymbol{S} \boldsymbol{D}_\Pi \boldsymbol{v} = \boldsymbol{D}_\Pi \boldsymbol{S} \boldsymbol{v} = Kp_t p_n \boldsymbol{v}.$$

Therefore, $\boldsymbol{v} = \text{vec}(\boldsymbol{\Pi})$ is an eigenvector of $\boldsymbol{B}$ with eigenvalue $\lambda_3 = Kp_t p_n$.

In summary, we identify the distinct eigenvalues of the matrix $\boldsymbol{B}$ as defined in (56) as follows:

$$\lambda_0 = 0, \quad \lambda_1 = p_n, \quad \lambda_2 = (1 - p_t + p_n) \cdot \frac{p_n + (K-1)p_t}{K}, \quad \lambda_3 = Kp_n p_t \tag{58}$$

with the corresponding multiplicities $2K - 1$, $K^2 - 3K + 1$, $K - 1$, and $1$, respectively.

Since the matrix $\boldsymbol{B}$ has zero eigenvalues, we consider its condition number within the non-zero eigenvalue subspace, which is given by $\lambda_3/\lambda_1 = Kp_t$. Consequently, we straightforwardly determine the condition number of the Hessian $\frac{\partial^2 \phi}{\partial \boldsymbol{w}^2}$ as defined in (52) is

$$\kappa(\nabla_{\boldsymbol{w}}^2 \phi) = Kp_t. \tag{59}$$

From the formula of $p_t$ given in (43), it is evident that increasing $\delta$ leads to a decrease of $p_t$, and consequently a reduction in the condition number.

By combining (51) and (59), we demonstrate that the Hessian matrices, $\nabla_{\boldsymbol{W}}^2 \phi(\boldsymbol{W}, \boldsymbol{H})$ and $\nabla_{\boldsymbol{H}}^2 \phi(\boldsymbol{W}, \boldsymbol{H})$, are positive semi-definite at the global optimizer. Moreover, the condition numbers of these Hessian matrices are notably lower under the label smoothing loss (with $0 < \delta < 1 - \sqrt{KN\lambda_W\lambda_H}$) compared to the cross-entropy loss (with $\delta = 0$). This observation suggests that the optimization landscape of the empirical loss function is better conditioned around its global minimizer under label smoothing loss, thus completing our proof.

$\square$

