# OpenReview forum: "Cross Entropy versus Label Smoothing: A Neural Collapse Perspective"
_TMLR — Accepted by TMLR_

### Review · Reviewer_owR4 · 2025-01-26

**Summary Of Contributions:**

The paper highlights the advantages of Label Smoothing loss over Cross-Entropy loss in deep neural networks. It demonstrates that LS loss accelerates convergence and improves generalization through enhanced Neural Collapse metrics. Theoretical insights show LS loss leads to a better optimization landscape. Additionally, it addresses LS loss's effects on model calibration, balancing improvements with potential overconfidence.

**Audience:**

Yes

**Claims And Evidence:**

Yes

**Requested Changes:**

Please check the weakness part above

**Strengths And Weaknesses:**

Strengths:

- The paper provides a fresh perspective on label smoothing by analyzing it through the framework of NC, which adds theoretical depth to the discussion.

- The paper presents empirical results demonstrating that models trained with label smoothing converge faster and achieve better generalization compared to those trained with cross-entropy loss. It also derives closed-form solutions for global minimizers under both loss functions

Weaknesses:

- The paper presents a theoretical analysis demonstrating that label smoothing leads to a lower conditioning number. However, it lacks an intuitive explanation for why this phenomenon occurs. Providing such insights would greatly enhance the understanding

- The impact of the smoothing hyperparameter $\delta$ on performance and calibration is mentioned, but further investigation into optimal $\delta$ values and their implications could provide clearer guidance for practitioners.

- The experiments are primarily conducted on relatively simple/specific datasets such as CIFAR-10, and STL-10. Including a broader range of datasets could enhance the generalizability of the findings.

---

> ### Author Response · Authors · 2025-03-07
>
> We thank you for your insightful and constructive feedback and for the positive comments you made regarding our work.   We fully agree that providing an intuitive explanation of label smoothing's impact and incorporating a broader range of experiments would improve the clarity and comprehensibility of our work.
>
> 1.  We agree that providing an intuitive explanation for why label smoothing leads to a lower condition number would enhance the understanding. In response, we have added an explanation at the end of Section 4.2. Specifically, without label smoothing, the cross-entropy loss encourages overconfident predictions, where nearly all probability mass is assigned to the target class, and non-target probabilities are pushed close to zero. This creates sharp curvature in dimensions corresponding to the target class and flat regions in non-target dimensions, resulting in a poorly conditioned optimization landscape with high curvature variations. Label smoothing mitigates this issue by encouraging a more uniform probability distribution, which stabilizes gradient updates and improves convergence efficiency.
>
> 2. Regarding the impact of the smoothing hyperparameter $\delta$ on calibration, our analysis focuses on how feature and weight vector norms, as well as Neural Collapse (NC1), influence calibration. From this perspective, we explain why label smoothing with a properly tuned $\delta$ improves calibration but may lead to higher Expected Calibration Error (ECE) if post-training temperature scaling is applied. However, it is important to recognize that calibration is influenced by many other factors, including test accuracy, and disentangling all these factors to provide a clear guideline for practitioners is challenging. While we acknowledge the importance of this issue, we leave a more comprehensive investigation of optimal $\delta$ values and their implications for future work.
>
> 3. We appreciate the suggestion to expand our experimental evaluation. While training on the full ImageNet dataset would be prohibitively expensive, we have incorporated Tiny ImageNet into our experiments, which shows similar trends as the other three datasets.

---

### Review · Reviewer_85QS · 2025-02-16

**Summary Of Contributions:**

This work provides a comprehensive study of the label smoothing loss from a neural collapse perspective. Empirically, the authors observe that compared to the cross-entropy loss, models trained with the label smoothing loss exhibit greater levels of NC1 (feature collapse) and NC2 (ETF structure), the latter explained by the fact that label smoothing equalizes logits not belonging to the target class. (Observations about ECE). Finally, the authors provide theoretical results that (1) characterize the global solution of the label smoothing loss, and (2) demonstrate that the Hessian condition number around the global solution is lower for the label smoothing loss compared to the cross-entropy loss.

**Audience:**

Yes

**Claims And Evidence:**

No

**Requested Changes:**

- As mentioned in the weaknesses, some (not all) of the claims are not extremely well-supported by the experimental results, which is why **I currently indicate below that the claims and evidence are not supported. However, if the authors could clarify or provide further results, I will change this.**
- The connection between label smoothing and calibration can be made more clear. I do understand how label smoothing can play the role of choosing the appropriate temperature by scaling the weight/feature norms. However, I’m confused by why the post temperature calibration is worse with larger label smoothing parameter, wouldn’t the temperature be selected to counteract the increase in delta? If the authors could elaborate on this it would be easier to appreciate.
- I think Theorem 4.2 can be improved by explicitly stating the decrease in the condition number for a given delta. As presented, one could choose delta to be very close to 0, which wouldn’t be much different from usual cross-entropy, but the theorem would still say this decreases the condition number and does not differentiate from any other delta in the given range.
- Certain figures (such as Figure 3, 4) have overlap in the legend and axes (e.g., Optimal T and Best T) which is confusing. If the quantity is specified in the axes, it would be better to not repeat in legend (or vice versa).
- Citations should be fixed to use parentheses form appropriately.
- Typo in caption of Figure 3, “optical”.

**Strengths And Weaknesses:**

Strengths:
- The paper is well-written and organized. Figures are well-formatted and professional.
- The finding that label smoothing has greater NC1 and NC2 is interesting. In particular, the explanation of more pronounced NC2 with label smoothing is intuitive and clear.
- Several practical implications can potentially be drawn from this work, such as tuning the smooth parameter to optimize model calibration and landscape.

Weaknesses:
- The authors argue that label smoothing can achieve better generalization due to improved NC2. While I find the connection between label smoothing and NC2 convincing, the generalization argument is not, particularly since most of the figures suggest there is little to no difference in error between label smoothing and cross entropy. The only result I could find that suggests otherwise was tucked away in the supplement in Figure 5 for STL10.
- Similarly, the argument that label smoothing leads to faster convergence also seems unsubstantiated. The provided figures show that the NC metrics converge faster for label smoothing, but this is not the same as the loss/error decreasing faster which is what is generally referred to for convergence.
- The theoretical result regarding the global solution is incremental, seemingly only providing an explicit value for the norm of weights/features compared to previous work.

---

> ### Author Response · Authors · 2025-03-07
>
> Thank you for your constructive and detailed feedback. We have carefully addressed your concerns as follows:
>
> 1. We acknowledge that the causal relationship between improved NC2 and model generalization is not firmly established. To address this, we have revised the last paragraph of Section 3.2 to provide a more precise explanation. Specifically, the convergence of NC2 to zero implies that the features and classifier weights converge to a simplex ETF structure. This geometric structure is inherently more robust to outliers and can theoretically enhance generalization. However, the impact of NC2 is intertwined with other factors, such as NC1. Due to these confounding factors, our current experiments cannot establish a direct causal relationship between NC2 and model generalization. We acknowledge this limitation and emphasize the need for further investigation to disentangle these effects.
>
> 2. Regarding the claim that label smoothing leads to faster convergence, we recognize that the loss formulations for models trained with cross-entropy and label smoothing differ, making direct comparisons of training loss not meaningful. In our original paper, we provided evidence of faster convergence based on NC metrics and the training error rate instead. To offer a clearer comparison, we have added Table 1, which presents the error rates at different epochs for models trained with and without label smoothing. Additionally, we also include a plot of log-scaled training loss for models trained under both settings in Figure 5 and Table 2 of the Appendix, which demonstrates that models trained with label smoothing converge faster in terms of training loss too.
>
> 3. Thank you for the suggestion on Theorem 4.2.  We have revised it to include the formula for the condition number as a function of the smoothing hyperparameter $\delta$, which shows that increasing $\delta$ results in a smaller condition number. (this was originally given in Section D in the appendix).  Additionally, we have included an intuitive explanation at the end of Section 4.2. Specifically, without label smoothing, cross-entropy loss drives overconfident predictions, leading to sharp curvature variations and a poorly conditioned optimization landscape. Label smoothing mitigates this by encouraging a more uniform probability distribution, stabilizing gradient updates, and improving convergence efficiency.
>
> 4. We examine the impact of label smoothing on model calibration through two main effects: (1) it regularizes the norms of feature embeddings and weight vectors, which has a similar effect as temperature scaling; (2) it significantly decreases NC1. When NC1 becomes too small, feature embeddings within a class become more concentrated, reducing the margin between correct and incorrect predictions. This can lead the model to be overconfident in its incorrect predictions, even after temperature scaling. Since temperature scaling uniformly rescales logits, it does not address the structural changes in feature embeddings caused by excessively small NC1. This explains why, when temperature scaling is applied to counter the regularization effect on feature/weight norms, label smoothing results in higher ECE, as it produces smaller NC1 compared to cross-entropy loss.
>
> 5. Thank you for pointing out the formatting issues and typos. We have also addressed these minor issues, including typos and citations.

---

### Review · Reviewer_LSeV · 2025-02-20

**Summary Of Contributions:**

This paper studies label smoothing, a pre-processing technique in training deep neural network (DNN) classifiers. It has been empirically shown that this procedure often leads to DNN models which generalize better than traditional one-hot encoding with cross-entropy loss. This paper focuses specifically on how label smoothing influences neural collapse and the implications this has on our theoretical understanding of DNNs. Empirically, it is shown that DNNs trained on smoothed labels exhibit neural collapse (NC1, NC2 & NC3) faster than DNNs trained on one-hot encoded labels. The paper also presents evidence that for the same level of NC1, models trained on smoothed labels exhibit intensified NC2. A short analysis on how the loss is effected by smoothed labels provides insight for why this might be the case. The paper also gives insight into how enhanced NC1 effects model calibration. Theoretically, under the unconstrained feature model, the paper presents closed form expressions for the global minimizers under both label smoothing and cross-entropy loss. Finally, it proves that label smoothing results in a lower condition number for the output weight matrix which could explain the faster rate of convergence.

**Audience:**

Yes

**Broader Impact Concerns:**

None.

**Claims And Evidence:**

Yes

**Requested Changes:**

These were also brought up under the weaknesses section, but summarizing here.

- Clearer evidence that LS exhibits faster convergence in both training and test errors. Possibly with a table of the error rate at different epochs instead of a plot.
- Clarify the analysis at the end of Section 3.2
- Quantify the improvement in the condition number when using LS over one-hot encoding in Theorem 4.2.
- Terminology of "LS loss".

**Strengths And Weaknesses:**

**Strengths**

- The neural collapse perspective is useful to the community and provides a new perspective on how to analyze label smoothing.
- The theoretical finding that label smoothing leads to a lower condition number in the solution is also interesting and potentially useful for developing the theory of optimization with DNNs.
- The paper is very well written and easy to follow. It also provides both empirical and theoretical support for the claims being made.

**Weaknesses**
- In Figure 1, the paper claims that LS exhibits faster convergence for both training and test errors, but these differences seem very minor. Is there a way to see the difference more clearly? Maybe with a table of the error rate at different epochs?
- The analysis at the end of Section 3.2 could be made more clear. My understanding is that the second equation is (almost) showing that
$$
\frac{\delta}{K} \sum_{l \neq k} \log(p_l)
$$
achieves its maximum when all $p_1 = p_{l'}$ for all $l,l' \neq k$. But the two terms aren't exactly the same so some more discussion or clarity on this would be appreciated.
- While the discussion on model calibration is nice, I feel it distracts a bit from the main message of the paper. To me it seems that the central message is about understanding how LS accelerates learning an NC model. I'm not suggesting the authors must remove this, but it was a bit distracting to me.
- Theorem 4.2 is unsatisfying. While it does say that LS will result in a better conditioned optimization problem is gives no insight into how much better. Some discussion on the improvement that you get with LS would improve the result.
-  The terminology "LS loss" is confusing. The paper is comparing label smoothing (which is a preprocessing of the data labels) with cross-entropy (the loss function used to train). Based on section 2.2 I believe it would make more sense to claim that this works compares **one-hot encoded labels with smoothed labels and applying cross-entropy loss in both cases**. Label smoothing is really about you're preprocessing the data not about changing the loss.

**Minor Comments/Typos**
- In Fig. 3 "optical T" should be "optimal T"
- In the introduction, when describing the "LS loss" the paper state that "label smoothing introduces a soft target label by blending the hard target label with a uniform distribution...". The word "blending" is quite vague and not necessary. It would be more clear to just say "LS preprocesses the label for each sample by taking a convex combination of the one-hot label and a uniform distribution."

---

> ### Author Response · Authors · 2025-03-07
>
> We thank you for your constructive and detailed feedback, which has really helped us improve the clarity and rigor of our work. We have addressed your concerns as follows:
>
> 1. We agree that label smoothing is really about how we preprocess the data rather than changing the loss function itself. We have revised the terminology throughout the paper to reflect this distinction.
>
> 2. To address your concern regarding the impact of label smoothing on training dynamics, we have added Table 1 in the main paper, which summarizes the training and testing errors during the early stages of training. The results clearly illustrate that models trained with label smoothing exhibit faster convergence in terms of training error.
>
> 3. Regarding Section 3.2, we have revised the equation to improve the clarity of the transition, as now shown in Equation (8).
>
> 4. With respect to the condition number, it was originally included in the proof in Section D of the Appendix. To improve clarity, we have now provided the formula for the condition number, which is a function of the smoothing hyperparameter $\delta$, in Theorem 4.2. This formula demonstrates that increasing $\delta$ results in a smaller condition number. Additionally, we have included an intuitive explanation at the end of Section 4.2. Specifically, without label smoothing, cross-entropy loss encourages overconfident predictions, where nearly all probability mass is assigned to the target class, and non-target probabilities are pushed close to zero. This creates sharp curvature in the target class dimensions and flat regions in the non-target dimensions, leading to a poorly conditioned optimization landscape with high curvature variations. Label smoothing mitigates this by promoting a more uniform probability distribution, stabilizing gradient updates, and improving convergence efficiency.
>
> Thank you again for your valuable suggestions.

---

### Decision · Action_Editor_eVAu · 2025-04-06

**Recommendation:** Accept as is

**Comment:**

The paper provides valuable insights into how label smoothing influences Neural Collapse, supported by rigorous empirical evidence and robust theoretical analysis. All reviewers agree that the paper is worth publication at TMLR.

Please incorporate all the comments from the reviewers into the paper for the final camera-ready.

**Audience:**

The study is likely to be of interest to the deep learning community, bridging theoretical insights and practical implications, and could inspire further research in understanding labeling smooth and feature learning.

**Claims And Evidence:**

This paper investigates label smoothing, a widely-used preprocessing technique in training deep neural network (DNN) classifiers, known empirically to improve generalization compared to traditional one-hot encoding combined with cross-entropy loss. Specifically, the study examines how label smoothing affects neural collapse (NC) and explores the theoretical understanding of feature learning in DNN.

Empirical results of this work demonstrate that DNNs trained with label smoothing reach states of neural collapse (NC1, NC2, and NC3) more rapidly than those trained with one-hot encoded labels. Furthermore, the findings reveal that for equivalent levels of NC1, models trained with smoothed labels exhibit more salient NC2. A brief analysis of the influence of label smoothing on loss dynamics offers insights into why this phenomenon occurs. Additionally, the study provides insights into how NC1 impacts model calibration.

From a theoretical perspective, the authors derive closed-form expressions for global minimizers under the unconstrained feature model for both label smoothing and standard cross-entropy loss. They further establish that label smoothing yields a lower condition number for the output weight matrix, potentially explaining the observed acceleration in convergence rates.

The paper provides valuable insights into how label smoothing influences Neural Collapse, supported by rigorous empirical evidence and robust theoretical analysis.